# The Geometric Reasoner: Manifold-Informed Latent Foresight Search for Long-Context Reasoning

**Ren Zhuang**[1]  **Ben Wang**[1]  **Shuifa Sun**[1]

## Abstract

Scaling test-time compute enhances long chain-of-thought (CoT) reasoning, yet existing approaches face a fundamental trade-off between computational cost and coverage quality: either incurring high training expense or yielding redundant trajectories. We introduce The Geometric Reasoner (TGR), a training-free framework that performs manifold-informed latent foresight search under strict memory bounds. At each chunk boundary, TGR scores candidate latent anchors via a lightweight look-ahead estimate combined with soft geometric regularizers that encourage smooth trajectories and diverse exploration. Chunk-wise KV cache resets keep memory linear in chunk length. On challenging math and code benchmarks, TGR improves robust trajectory coverage, measured by the area under the Pass@$k$ curve (AUC), by up to 13 points on Qwen3-8B, with negligible overhead of about 1.1–1.3×.

## 1. Introduction

Scaling test-time compute reliably enhances large language model (LLM) performance, unlocking reasoning capabilities previously out of reach (Xu et al., 2025; Muennighoff et al., 2025; Zhang et al., 2025b; Wu et al., 2025b). Yet turning extra compute into diverse exploration without incurring prohibitive memory costs or wasting budget on redundant, correlated trajectories remains difficult. The quadratic complexity of attention and the footprint of KV caching (Vaswani et al., 2017; Beltagy et al., 2020; Dao et al., 2022; Dao, 2023) bottleneck practical deployment, rendering naive long-horizon exploration intractable.

Existing approaches make progress on this tension, yet criti-

[1]School of Information Science and Technology, Hangzhou Normal University, Hangzhou, China. Correspondence to: Ben Wang <20170056@hznu.edu.cn>, Shuifa Sun <watersun@hznu.edu.cn>.

*Proceedings of the $43^{rd}$ International Conference on Machine Learning*, Seoul, South Korea. PMLR 306, 2026. Copyright 2026 by the author(s).

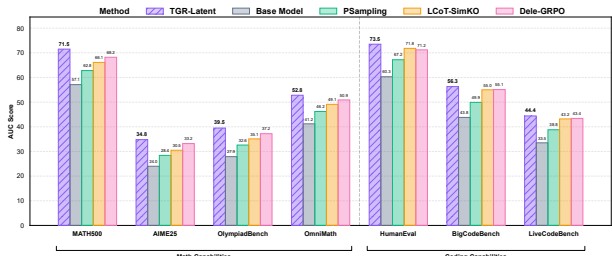

*Figure 1.* **TGR-Latent consistently outperforms baselines on Qwen3-8B.** Manifold-informed latent foresight search steering converts modest inference-time compute into robust coverage without weight updates.

cal gaps persist. Reinforcement Learning (RL) and preference optimization amortize long-horizon control into model weights (Schulman et al., 2017; Christiano et al., 2017; Shao et al., 2024; Zheng et al., 2025). While effective for single-sample accuracy, these methods demand substantial training compute and can collapse the trajectory distribution (Yue et al., 2025; Srivastava & Aggarwal, 2025). Sampling-based inference trades compute for robust coverage without retraining (Wang et al., 2022; Karan & Du, 2025; Faria et al., 2024), yet returns diminish rapidly as additional samples often remain correlated and fail to expand the set of distinct, valid trajectories (Tang et al., 2025; Wu et al., 2025b). These limitations motivate a training-free inference-time search procedure that allocates compute dynamically under strict memory bounds.

To address this, we introduce **The Geometric Reasoner (TGR)**, a training-free framework for manifold-informed latent foresight search. TGR formulates reasoning as path search on a latent manifold but avoids rigid constraints that become brittle in high dimensions (Yallup et al., 2025; Betancourt, 2017; Goyal & Shetty, 2019). Instead, it ranks candidate reasoning chunks via a soft geometric score combining lightweight look-ahead utility, a bumpiness penalty for smooth transitions, and a repulsive diversity force that discourages similarity. Chunk-wise KV cache resets keep memory linear in chunk length, enabling scalable long-context exploration.

This chunk-boundary state interface relates to markovian

control approaches that learn explicit state transitions, like Delethink (Aghajohari et al., 2025). Rather than relying on learned state transitions or external verifiers, TGR enables training-free inference-time search, allocating compute to achieve controllable exploration and robust coverage under matched budgets.

Our contributions are:

- **Manifold-Informed Latent Foresight Search.** We propose TGR, a training-free inference-time framework that scores chunk-level latent anchors with a soft geometric objective, enabling controllable exploration without parameter updates.

- **Budget-Efficient Robustness.** TGR improves robust trajectory coverage on challenging math and code benchmarks, achieving up to 13-point AUC (area under the Pass@$k$ curve) gains on Qwen3-8B with negligible overhead of about 1.1–1.3×, as shown in Figure 1.

- **Scaling Limits of Hard Geometry.** We demonstrate that hard geometric filtering suffers from vanishing feasibility in high-dimensional latent spaces, where the acceptance rate decays exponentially as dimension increases, establishing soft geometric regularization as a scalable alternative.

**Conflict of Interest Disclosure.** The authors declare no financial or other substantive conflicts of interest.

## 2. Related Work

### 2.1. Test-Time Reasoning

Prompt engineering steers model behavior via input formatting and exemplars, spanning in-context learning (Brown et al., 2020), Chain-of-Thought decomposition (Wei et al., 2022; Kojima et al., 2022), and program-aided methods such as PAL (Gao et al., 2023). These approaches improve controllability without additional training but remain sensitive to prompt design and operate via token-level heuristics.

Sampling-based reasoning includes local decoding controls (temperature, top-$k$, top-$p$), candidate pooling with Best-of-$N$ selection (Wan et al., 2025), and aggregation methods such as Self-Consistency (Wang et al., 2022). Aggressive test-time sampling shows that additional inference compute can elicit latent reasoning without retraining (Karan & Du, 2025). Approaches such as Tree of Thoughts (ToT) (Yao et al., 2023) and Graph of Thoughts (GoT) (Besta et al., 2024) exemplify structured inference, which introduces explicit search in token space and trades computation for robustness. However, these methods search over discrete continuations and rely on token-space proxies or external scoring, rendering long-horizon search brittle under limited budgets.

### 2.2. Reinforcement Learning and Preference Optimization

Amortizing long-horizon control into model parameters via reinforcement learning and preference optimization has become a dominant paradigm (Christiano et al., 2017; Zhang et al., 2025a; Wiering & Van Otterlo, 2012). Beyond PPO (Schulman et al., 2017) and DPO (Rafailov et al., 2023), recent methods including GRPO (Shao et al., 2024) and GSPO (Zheng et al., 2025) scale reasoning by reducing reliance on a learned critic. Delethink (Aghajohari et al., 2025) learns discrete state transitions, demonstrating the utility of explicit Markovian states for long-horizon control. While effective, reward-and-preference-driven training exhibits failure modes at reduced diversity and overconfident generations (Yue et al., 2025; Srivastava & Aggarwal, 2025). SimKO (Peng et al., 2025) mitigates mode collapse and length bias, yet these training-centric approaches remain computationally intensive.

### 2.3. Geometric Perspectives on Latent Modeling

Geometric Deep Learning interprets learned representations as structured spaces (Bronstein et al., 2017; Higgins et al., 2017). Recent works frame attention as a geometric object governing representation transport along a curved manifold, treating layers as discrete geodesic steps shaped by training (Aggarwal et al., 2025; Di Sipio et al., 2025; Zhu et al., 2025). Complementing this view, LLM decoding is interpreted as tracing trajectories on a low-dimensional manifold, where coherent generations follow smooth paths and hallucinations appear as abrupt jumps (Zhang & Dong, 2025). These insights motivate geometric inductive biases at training time: manifold priors (Kang et al., 2025) and architectural constraints such as mHC (Xie et al., 2025), which applies Sinkhorn–Knopp normalization (Sinkhorn & Knopp, 1967) to confine signal propagation to a stable low-dimensional structure. Soft enforcement of geometry is preferred over hard feasibility filtering, which becomes impractical in high dimensions due to concentration of measure (Yallup et al., 2025; Betancourt, 2017; Goyal & Shetty, 2019). TGR operationalizes this insight at inference time, translating geometric priors into differentiable soft penalties without architectural changes.

Figure 2 contrasts TGR with test-time reasoning and RL methods, highlighting its training-free, manifold-informed latent foresight search.

## 3. Method

TGR is a training-free geometric reasoner that operates in latent space. It keeps the backbone LLM frozen, segments reasoning into chunks, and performs search at chunk boundaries within a continuous anchor space. At each boundary,

*Figure 2.* **Overview of reasoning frameworks.** Unlike **(a) test-time sampling**, exploring trajectories without explicit structure, or **(b) reinforcement learning**, which internalizes preferences through costly training, **(c) TGR** introduces a training-free inference-time search over the latent manifold. It selects optimal chunk-level anchors via a soft geometric score combining foresight, bumpiness, and uniformity, then injects them into generation with KV cache resets to maintain linear memory cost.

TGR samples candidate anchors, scores them using computationally efficient rollouts and soft geometric penalties, and selects the highest-scoring anchor to steer the subsequent chunk. Note that in this work, TGR denotes the latent-space variant TGR-Latent, unless explicitly stated otherwise.

## 3.1. Latent Space State Representation

Given a query $Q$, TGR generates output as a sequence of $L$ chunks $\tau_{1:L} = (\tau_1, \ldots, \tau_L)$, each of maximum length $S$. Let $c_{t-1}$ denote the token context preceding chunk $t$. To bound memory during long-horizon decoding, we employ chunk-wise KV cache resets: at each boundary, we rebuild the prefix cache from $c_{t-1}$, discarding earlier history while transmitting long-range state via latent anchors (Appendix A). Concretely, $c_{t-1}$ comprises the query and recent token suffix, left-truncated to maximum length $S$.

### 3.1.1. STATE ANCHORS

We represent chunk-level reasoning states as unit-norm latent anchors on $\mathcal{Z} \triangleq \{x \in \mathbb{R}^{d_z} : \|x\|_2 = 1\}$. A state anchor $z_t \in \mathcal{Z}$ is extracted upon completing chunk $t$, serving as a compact chunk-boundary summary. Unlike Delethink (Aghajohari et al., 2025), which learns discrete state transitions, TGR extracts anchors directly from hidden states and performs score-guided anchor selection at inference time. This provides a summary for control under bounded KV cache constraints. At the next boundary, we sample $K$ candidate anchors $\{\hat{z}_t^{(j)}\}_{j=1}^K \subset \mathcal{Z}$ centered at $z_{t-1}$, evaluate each via lightweight rollouts, and select the highest-scoring candidate to steer the next chunk.

This design addresses two inference constraints: chunk-wise KV resets require explicit state summaries for long-horizon coherence, and effective compute allocation demands a stateful interface rather than unstructured sampling.

### 3.1.2. STATE EXTRACTION

Upon generating $\tau_t$, we append a fixed end-of-chunk delimiter token $<\text{EOC}>$ and project its top-layer hidden state $h_{\text{eoc}}(\tau_t) \in \mathbb{R}^{d_h}$ onto the unit sphere , equivalently, the last hidden state of $\tau_t \oplus <\text{EOC}>$:

$$z_t \triangleq \frac{W h_{\text{eoc}}(\tau_t)}{\|W h_{\text{eoc}}(\tau_t)\|_2} \in \mathcal{Z}, \tag{1}$$

where $W \in \mathbb{R}^{d_z \times d_h}$ is a fixed projection matrix. This yields a compact, model-internal summary facilitating chunk-to-chunk state transfer despite context truncation.

## 3.2. Foresight Search

We generate candidate anchors by sampling from the local neighborhood of the current state anchor via tangent-space perturbation followed by re-normalization:

$$\begin{aligned}
\epsilon &\sim \mathcal{N}(0, I), \\
v &\leftarrow \epsilon - (z^\top \epsilon)z, \\
\tilde{a} &\leftarrow z + \sigma v, \\
a &\leftarrow \frac{\tilde{a}}{\|\tilde{a}\|_2} \in \mathcal{Z},
\end{aligned} \tag{2}$$

where $z$ is the center anchor, setting $z = z_{t-1}$, and $\sigma$ controls the exploration radius. Tangent-space perturbations preserve the local neighborhood structure on the unit sphere, avoiding radial drift associated with ambient space rescaling. We sample $K$ candidate anchors $\{\hat{z}_t^{(j)}\}_{j=1}^K$ around $z_{t-1}$ using Eq. (2). Each candidate is evaluated using the manifold-informed soft geometric score (Section 3.3), and the optimal anchor is selected to generate the subsequent chunk.

## 3.3. Manifold-Informed Regularization

We operationalize soft geometry as manifold-informed regularization that exploits local structure to guide search, leveraging tangent neighborhoods and curvature proxies. Unlike hard feasibility filtering, our approach translates geometric priors into differentiable soft penalties.

At each boundary, we score a candidate anchor $a$ given context $c$ and previous anchor $z$ by maximizing:

$$\text{Score}(a; c, z) = V_{\text{fore}}(a; c) \\ - \lambda_b \, P_{\text{bum}}(a; c) - \lambda_u \, P_{\text{uni}}(a; z). \quad (3)$$

Here, $\lambda_b, \lambda_u \geq 0$ balance capability, coherence, and exploration.

### 3.3.1. FORESIGHT VALUE.

We estimate look-ahead value via a low-depth rollout of $s \ll S$ tokens under $p_\theta(\cdot \mid c, a)$, where $p_\theta$ denotes the frozen base model conditioned on anchor $a$ via residual injection:

$$V_{\text{fore}}(a; c) \triangleq \mathbb{E}\left[\frac{1}{s}\sum_{i=1}^{s}\log p_\theta(\hat{\tau}_i \mid \hat{\tau}_{<i}; c, a)\right], \\ \hat{\tau}_{1:s} \sim p_\theta(\cdot \mid c, a). \quad (4)$$

We approximate this expectation with a single Monte Carlo rollout per candidate. In this setting, $V_{\text{fore}}$ becomes a single-sample average log-likelihood proxy sufficient for ranking. This direct measurement of consistency in low-cost rollouts enables ranking without external reward models or trained verifiers.

### 3.3.2. BUMPINESS PENALTY.

Let $\mathbf{g}_1, \ldots, \mathbf{g}_s$ be the top-layer hidden states along the rollout. We penalize high-frequency variation in the latent trajectory via a discrete second-order difference:

$$P_{\text{bum}}(a; c) \triangleq \frac{1}{s-2}\sum_{i=2}^{s-1}\|\mathbf{g}_{i+1} - 2\mathbf{g}_i + \mathbf{g}_{i-1}\|_2^2. \quad (5)$$

This term discourages abrupt directional changes, functioning as an efficient proxy for cross-chunk coherence.

### 3.3.3. UNIFORMITY REGULARIZER.

Since $a, z \in \mathcal{Z}$ are normalized, their semantic similarity reduces to a dot product. We enforce uniformity through repulsive interactions via a hinge penalty with threshold $\delta \in [0, 1)$, thereby promoting diverse exploration:

$$P_{\text{uni}}(a; z) \triangleq \max\{0, \, a^\top z - \delta\}. \quad (6)$$

This penalizes minimal-drift updates only when similarity exceeds $\delta$ (default $\delta = 0.2$, see Appendix C.8).

---

**Algorithm 1** TGR Reasoning Process

**Require:** Query $Q$, chunk limit $L$, chunk length $S$, candidate anchors $K$, rollout length $s$.
**Require:** Proposal radius $\sigma$, weights $\lambda_b, \lambda_u$, threshold $\delta$.
**Require:** Frozen base model parameters $\theta$; fixed projection $W$ defined in Eq. 1; fixed injectors $\{A_\ell, B_\ell\}$ defined in Eq. 8.
**Ensure:** Output trajectory $\tau_{1:T}$.
1: Initialize context $c_0 \leftarrow \text{Window}(Q)$.
2: Run frozen model on $c_0$, set $z_0 \leftarrow \text{Normalize}(W h_{\text{eoc}}(c_0))$ via Eq. 1.
3: **for** $t = 1, \ldots, L$ **do**
4:     Rebuild prefix KV cache for $c_{t-1}$ via chunk-wise reset.
5:     **for** $j = 1, \ldots, K$ **do**
6:         Sample candidate $\hat{z}_t^{(j)} \leftarrow \text{SAMPLEAROUND}(z_{t-1}, \sigma)$ using Eq. 2.
7:         Roll out $\hat{\tau}_{1:s}^{(j)} \sim p_\theta(\cdot \mid c_{t-1}, \hat{z}_t^{(j)})$ for $s$ steps, applying residual injection via Eq. 8.
8:         Compute foresight value $V_{\text{fore}}(\hat{z}_t^{(j)}; c_{t-1})$ via Eq. 4.
9:         Compute bumpiness $P_{\text{bum}}(\hat{z}_t^{(j)}; c_{t-1})$ via Eq. 5.
10:        Compute uniformity $P_{\text{uni}}(\hat{z}_t^{(j)}; z_{t-1})$ via Eq. 6.
11:        Score $\mathcal{S}^{(j)} \leftarrow V_{\text{fore}} - \lambda_b P_{\text{bum}} - \lambda_u P_{\text{uni}}$ using Eq. 3.
12:     **end for**
13:     Select $j^\star \leftarrow \arg\max_j \mathcal{S}^{(j)}$; set $\hat{z}_t \leftarrow \hat{z}_t^{(j^\star)}$.
14:     Generate chunk $\tau_t \sim p_\theta(\cdot \mid c_{t-1}, \hat{z}_t)$, length $\leq S$, via injection as in Eq. 8.
15:     Extract new anchor $z_t \leftarrow \text{Normalize}(W h_{\text{eoc}}(\tau_t))$ via Eq. 1.
16:     Update context $c_t \leftarrow \text{Window}(Q \oplus \tau_t)$ resetting KV cache.
17:     **if** stopping criterion met **then**
18:         **break**
19:     **end if**
20: **end for**
21: **return** $\tau_{1:t}$

---

## 3.4. Overall Process

Algorithm 1 details the TGR inference process. At each boundary, we sample $K$ candidate anchors, compute the manifold-informed score using efficient rollouts, select the highest-scoring anchor, generate the next chunk, and update the state anchor. Formally, letting $\mathcal{A}_t \triangleq \{\hat{z}_t^{(j)}\}_{j=1}^{K}$ be the candidate set at step $t$, we select:

$$\hat{z}_t \triangleq \arg\max_{a \in \mathcal{A}_t} \text{Score}(a; c_{t-1}, z_{t-1}). \quad (7)$$

**Anchor Conditioning via Residual Injection** We condition on anchor $a_t \in \mathcal{Z}$ using a fixed low-rank residual injection at each transformer layer $\ell$:

$$\mathbf{h}_\ell \leftarrow \mathbf{h}_\ell + B_\ell A_\ell a_t, \\ A_\ell \in \mathbb{R}^{r \times d_z}, \quad B_\ell \in \mathbb{R}^{d_h \times r}, \quad r \ll d_h. \quad (8)$$

This creates a lightweight control interface without parameter updates. We inject at every layer using identical interfaces for candidate evaluation and generation, with fixed random matrices of rank $r$ (default $r = 8$) to maintain the training-free setting (Appendix A).

### 3.5. Vanishing Acceptance of Hard Geometric Constraints

A standard method for imposing latent geometric structure is hard feasibility filtering—accepting candidates only if they satisfy constraints such as geodesic-following, curvature caps, or angular thresholds.

Let $a$ denote a candidate sampled from proposal distribution $q(\cdot \mid z_{t-1})$ centered at $z_{t-1}$, and let $\mathcal{F}_t^{\text{geo}}$ be the feasible set induced by a hard constraint, namely a curvature-bounded geodesic-following rule. The feasibility acceptance rate, i.e., pure accept/reject excluding likelihood ratios, is:

$$
\begin{aligned}
\alpha &\triangleq \Pr_{a \sim q(\cdot \mid z_{t-1})}\left[a \in \mathcal{F}_t^{\text{geo}}\right] \\
&= \mathbb{E}_{a \sim q(\cdot \mid z_{t-1})}[\mathbb{I}\{a \in \mathcal{F}_t^{\text{geo}}\}].
\end{aligned}
\tag{9}
$$

The expected number of proposals required to obtain one feasible sample is $\mathbb{E}[N_{\text{prop}}] = 1/\alpha$. Maintaining a fixed number of feasible candidates, such as $K$ candidates per step, therefore requires inflating the proposal budget by a factor of roughly $1/\alpha$.

Crucially, $\alpha$ collapses rapidly in high-dimensional latent spaces due to concentration of measure; feasible regions often occupy an exponentially small fraction of the local neighborhood as dimensionality increases (Betancourt, 2017; Goyal & Shetty, 2019; Yallup et al., 2025). Table 6 illustrates that hard filtering becomes computationally prohibitive as dimension increases.

This motivates our use of soft geometry via manifold-informed regularization. Rather than rejecting candidates outside $\mathcal{F}_t^{\text{geo}}$, TGR integrates geometric priors as differentiable penalties within the scoring objective. This approach preserves geometric bias while circumventing the prohibitive $1/\alpha$ cost penalty of hard feasibility filtering.

## 4. Experiments

### 4.1. Experimental Setup

**Models**  We use Qwen3 (Yang et al., 2025) as the primary testbed and report main results for the 8B variant; additional scales, 1.7B, 4B and 14B, are deferred to Appendix D. To study the interaction with training stages, we also evaluate Phi-4-reasoning (SFT Pre-Training) and Phi-4-reasoning-plus (with RL Post-Training) (Abdin et al., 2025).

**Baselines**  We compare against (i) *training-free test-time compute* baselines: the base model and Power Sampling (PSampling) (Karan & Du, 2025); (ii) *RL-tuned approaches* that internalize long-CoT behaviors, including LCoT(GRPO) (Guo et al., 2025) and LCoT(SimKO) (Peng et al., 2025); (iii) *RL with structured state transitions*, instantiated by Delethink (Aghajohari et al., 2025) combined

with GRPO or SimKO; and (iv) *token-space control ablations*, which mirrors strong test-time strategies like Self-Consistency (Wang et al., 2022), ToT (Yao et al., 2023) or GoT (Besta et al., 2024) that operate in token-space under the same chunked interface.

**Benchmarks**  For mathematics, we use MATH500 (Hendrycks et al., 2021), AIME2025 (MAA, 2025), OmniMath (Gao et al., 2024), and OlympiadBench (He et al., 2024). For code generation, we use HumanEval (Chen, 2021), BigCodeBench (Zhuo et al., 2024), and LiveCodeBench (Jain et al., 2024).

**Unified Inference Budget and Metrics.**  We report Pass@$k$ for $k \in \mathcal{K}$ and the normalized area under the Pass@$k$ curve, namely AUC as robustness metrics, along with Avg. Tokens, defined as the average end-to-end tokens per problem required to produce $k$ final trajectories (Appendix C.3).

We define the normalized AUC on a log scale as

$$
\begin{aligned}
\text{AUC} \triangleq{} & \frac{100}{\log_2 k_M - \log_2 k_0} \sum_{i=0}^{M-1} \left( \log_2 k_{i+1} - \log_2 k_i \right) \\
& \cdot \frac{\text{Pass@}k_{i+1} + \text{Pass@}k_i}{2},
\end{aligned}
\tag{10}
$$

where $\mathcal{K} = \{k_0 < \cdots < k_M\}$. For code benchmarks, Pass@$k$ is computed using the unbiased estimator of Chen (2021). For math benchmarks, Pass@$k$ is the empirical success rate that at least one of $k$ samples is correct. We use temperature $0.6$ and the default TGR sampling with $K = 8$ candidate anchors per chunk boundary, scored by rapid rollouts ($s = 32$ on math tasks and $s = 64$ on code tasks); full hyperparameters are listed in Appendix C.8. We set an upper bound of $L = 24$ chunks with maximum chunk length $S = 512$, while actual generation terminates early by standard stopping criteria; all reported costs are based on realized token consumption.

### 4.2. Main Results

We evaluate TGR on math and code benchmarks under matched inference budgets. TGR-Latent consistently improves robust trajectory coverage while remaining token-efficient, indicating higher effective sample efficiency than both training-free sampling and training-heavy RL baselines. The comparison with TGR-Token isolates the benefit of latent-space search beyond discrete token-level control.

**Mathematical Reasoning**  Table 1 reports results on Qwen3-8B. TGR-Latent consistently improves robust coverage, with gains scaling at medium-to-large sampling budgets. This indicates effective conversion of samples into distinct solution trajectories. Moreover, TGR attains these

*Table 1.* **Mathematical reasoning results on Qwen3-8B.** TGR-Latent matches or exceeds the best baseline on AUC while consuming 18% fewer tokens, demonstrating that inference-time latent search achieves superior budget-efficiency without training updates.

| Method | AIME25 | | | | OlympiadBench | | | | OmniMath | | | | Tokens |
|---|---|---|---|---|---|---|---|---|---|---|---|---|---|
| | @1 | @32 | @128 | AUC | @1 | @32 | @128 | AUC | @1 | @32 | @128 | AUC | Avg. ($\times 10^3$) |
| Base Model | 18.7 | 26.5 | 28.4 | 24.0 | 22.1 | 30.2 | 32.6 | 27.9 | 31.4 | 45.8 | 48.9 | 41.2 | 2.6 |
| PSampling | 23.8 | 30.1 | 32.1 | 28.4 | 27.4 | 34.5 | 36.8 | 32.6 | 38.7 | 49.8 | 52.3 | 46.2 | 6.0 |
| LCoT-GRPO | 26.5 | 27.8 | 28.5 | 27.5 | 30.1 | 31.8 | 32.5 | 31.5 | 41.8 | 48.0 | 49.8 | 46.2 | 8.5 |
| LCoT-SimKO | 27.5 | 31.5 | 33.0 | 30.5 | 31.8 | 36.0 | 38.0 | 35.1 | 43.5 | 51.5 | 54.0 | 49.1 | 8.6 |
| Dele-GRPO | 27.4 | 35.5 | 37.8 | 33.2 | 31.5 | 39.0 | 41.8 | 37.2 | 43.1 | 54.2 | 57.1 | 50.9 | 8.4 |
| Dele-SimKO | **28.4** | 37.0 | 39.5 | **34.8** | 32.5 | 40.8 | 43.8 | 38.8 | **44.6** | 56.1 | 59.2 | **52.8** | 8.5 |
| TGR-Token | 25.4 | 33.2 | 35.6 | 31.1 | 29.1 | 37.0 | 39.5 | 34.8 | 41.2 | 53.0 | 55.8 | 49.3 | 7.5 |
| TGR-Latent | 27.8 | **37.4** | **40.1** | **34.8** | **32.8** | **41.8** | **44.9** | **39.5** | 43.8 | **56.8** | **60.1** | **52.8** | 7.0 |

*Table 2.* **Code generation results on Qwen3-8B.** TGR-Latent achieves the highest AUC on all three benchmarks and leads at high $k$, where latent-space diversity enables exploration of multiple valid coding solutions.

| Method | HumanEval | | | | BigCodeBench | | | | LiveCodeBench | | | | Tokens |
|---|---|---|---|---|---|---|---|---|---|---|---|---|---|
| | @1 | @32 | @128 | AUC | @1 | @32 | @128 | AUC | @1 | @32 | @128 | AUC | Avg. ($\times 10^3$) |
| Base Model | 45.7 | 67.8 | 71.2 | 60.3 | 34.1 | 48.5 | 51.3 | 43.8 | 26.8 | 36.8 | 38.9 | 33.5 | 1.6 |
| PSampling | 52.8 | 74.5 | 77.8 | 67.2 | 40.8 | 54.2 | 57.2 | 49.9 | 32.4 | 42.1 | 44.1 | 38.8 | 3.6 |
| LCoT-GRPO | 55.0 | 76.8 | 79.5 | 69.5 | 43.5 | 57.5 | 60.2 | 52.8 | 35.5 | 44.5 | 46.8 | 41.5 | 6.0 |
| LCoT-SimKO | 56.2 | 79.2 | 82.0 | 71.8 | 44.8 | 60.1 | 62.8 | 55.0 | 36.5 | 46.5 | 48.8 | 43.2 | 6.0 |
| Dele-GRPO | 56.4 | 79.8 | 82.5 | 71.2 | 44.8 | 60.5 | 63.3 | 55.1 | 36.4 | 46.8 | 49.1 | 43.4 | 5.9 |
| Dele-SimKO | **57.1** | 80.5 | 83.5 | 73.2 | 45.5 | 61.3 | 64.1 | 56.0 | **37.2** | 47.5 | 49.8 | 44.1 | 5.8 |
| TGR-Token | 55.3 | 78.5 | 81.8 | 70.8 | 43.5 | 58.8 | 61.4 | 53.6 | 35.1 | 45.5 | 47.8 | 42.0 | 4.4 |
| TGR-Latent | 56.1 | **81.1** | **84.2** | **73.5** | **45.6** | **62.1** | **64.8** | **56.3** | 36.8 | **48.1** | **50.2** | **44.4** | 4.6 |

gains with token costs competitive with training-heavy RL baselines.

**Code Generation** Table 2 summarizes code generation results. TGR-Latent improves AUC consistently while maintaining comparable token cost. The advantage is pronounced on coding tasks, which often admit multiple correct solutions. Substantial gains at higher $k$ values are achieved by TGR through effective exploration of diverse coding trajectories in latent space.

**Ablation of Geometric Components.** Table 3 isolates each score component. Removing look-ahead value $V_{fore}$ yields the largest degradation, confirming that look-ahead guidance from limited-horizon look-ahead rollouts is the primary driver. Removing uniformity $P_{uni}$ substantially reduces AUC, indicating that repulsion is essential for candidate pool diversity. Bumpiness $P_{bum}$ stabilizes multi-chunk coherence. The gap between TGR-Latent and TGR-Token suggests that improvements stem from latent-space search, not merely chunking.

**Robustness Checks.** To isolate chunk length from the total reasoning horizon, we vary $S \in \{256, 512, 1024\}$ and adjust $L$ so that $L \times S = 12288$ remains fixed. Table 4 shows a predictable adaptivity–efficiency trade-off rather

*Table 3.* **Component ablation on Qwen3-8B.** Lightweight foresight is essential for performance, while bumpiness and uniformity regularizers provide complementary gains. Latent-space control consistently outperforms its discrete token-level counterpart.

| Method | MATH500 | AIME25 | Avg. AUC | Δ |
|---|---|---|---|---|
| TGR-Latent | 71.5 | 34.8 | 53.2 | - |
| - w/o $P_{uni}$ | 67.2 | 31.0 | 49.1 | - 7.6% |
| - w/o $P_{bum}$ | 69.1 | 31.7 | 50.9 | - 4.2% |
| - w/o $V_{fore}$ | 61.3 | 26.1 | 43.7 | - 17.8% |
| - random anchor | 55.1 | 21.5 | 38.3 | - 27.9% |
| TGR-Token | 66.4 | 31.1 | 48.8 | - 8.3% |

than brittle sensitivity to $S$: smaller chunks improve control frequency at higher overhead, while larger chunks reduce overhead with mild coverage loss. The default $S = 512$ lies near the stable operating region. Additional token-space baselines and pilot open-ended evaluations are reported in Appendices D.2 and D.3.

### 4.3. In-Depth Analysis of TGR on Qwen3-8B

To understand TGR's geometric design and clarify the source of its cost–robustness advantage, we interpret how soft geometric scoring balances foresight, smoothness, and diversity to sustain effective coverage under budget constraints.

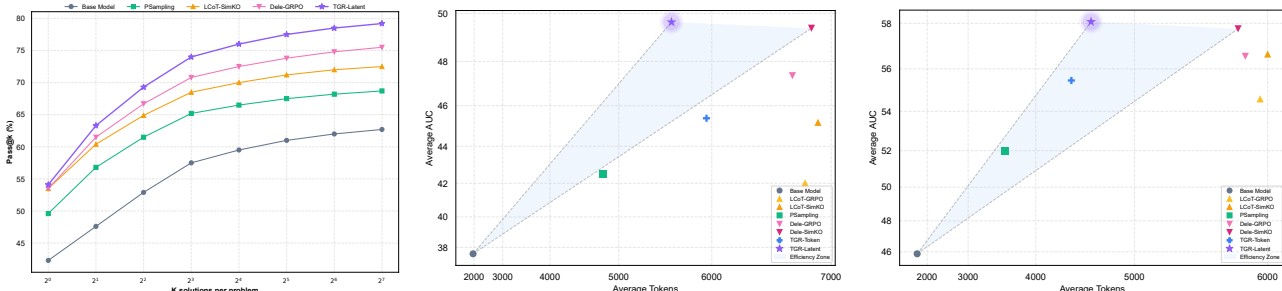

*Figure 3.* **TGR dominates the inference efficiency frontier. Left:** Pass@$k$ curves on MATH500 reveal that TGR-Latent sustains marginal gains beyond $k = 32$ where baselines plateau. **Middle & Right:** On the cost–robustness plane, TGR-Latent occupies the upper-left corner, achieving the highest AUC at moderate token cost on both math and code benchmarks.

*Table 4.* **Chunk-length sensitivity on Qwen3-8B.** With the total horizon fixed, TGR changes smoothly across chunk lengths rather than depending on a brittle value of $S$.

| Benchmark | Chunk length | AUC | Pass@128 | Overhead |
|---|---|---|---|---|
| | $S = 256$ | 35.2 | 40.7 | 1.17$\times$ |
| AIME25 | $S = 512$ (default) | 34.8 | 40.1 | 1.11$\times$ |
| | $S = 1024$ | 33.9 | 38.9 | 1.05$\times$ |
| | $S = 256$ | 73.7 | 84.5 | 1.16$\times$ |
| HumanEval | $S = 512$ (default) | 73.5 | 84.2 | 1.11$\times$ |
| | $S = 1024$ | 72.8 | 83.1 | 1.06$\times$ |

**Inference Scaling on MATH500.** Figure 3 (left) plots Pass@$k$ on MATH500. The critical signal is the marginal gain as $k$ increases. TGR-Latent sustains larger improvements at medium-to-large $k$, consistent with higher candidate independence. The repulsive term minimizes overlap, ensuring additional samples translate into new trajectory modes. Value-based look-ahead sharpens boundary selection, converting rollout compute into sustained gains.

**Cost–Robustness Frontier.** Figure 3 (middle, right) summarizes the cost–robustness trade-off. Methods relying on training-heavy policies can incur substantial token cost without proportional coverage increase, while sampling-based approaches saturate early. TGR-Latent makes the most of its compute budget, allocating it to lightweight look-ahead and pruning low-yield exploration. This supports the conclusion that soft geometric scoring improves the conversion rate from inference compute to trajectory coverage.

**Mode Collapse and Search Dynamics.** To diagnose diversity collapse, we analyze the geometry of within-step candidate anchors over time. As shown in Figure 4 (left), RL-tuned baselines concentrate probability mass in a narrow latent region, yielding highly correlated candidates and reduced coverage. In contrast, TGR maintains a dispersed anchor distribution, reflecting the effect of geometric repulsion in sustaining effective parallelism during scoring. This qualitative observation is quantified in Table 13, where removing $P_{\text{uni}}$ collapses the candidate set toward a narrow

cone, while the full model preserves meaningful separation.

**Hyperparameter Sensitivity.** Figure 4 (right) reports sensitivity to rollout depth $s$ and the candidate budget $K$, defined as the number of candidate anchors scored per boundary. Increasing either parameter improves AUC but with diminishing returns, while overhead grows approximately linearly. Consistent with the design goal of budget-aware inference-time search, where moderate look-ahead is sufficient to rank anchors reliably, and soft geometric penalties maintain diversity without excessive sampling, the default configuration lies near a stable operating region.

### 4.4. Interpreting TGR Representations

Beyond performance gains, we seek to interpret how TGR's geometric design shapes its internal representations and search behavior, shedding light on the mechanisms underlying robust coverage.

**Balancing Value, Bumpiness, and Uniformity in TGR.** Soft geometric scoring manages a three-way trade-off among look-ahead value, cross-chunk coherence, and exploration. Greedy decoding over-optimizes near-horizon likelihood; aggressive sampling sacrifices coherence. TGR makes this trade-off explicit via a single objective combining rollout value with bumpiness and uniformity regularizers. Regularization weights $(\lambda_b, \lambda_u)$ control coherence and coverage. Empirically, removing $V_{\text{fore}}$ degrades performance most, while removing $P_{\text{uni}}$ primarily reduces AUC, indicating diminished parallelism.

**Effective Independence Explains Stronger Gains in AUC than in Pass@1.** AUC aggregates performance across the sampling scale, proxying the effective number of distinct trajectories. When candidates are correlated, increasing $k$ yields diminishing returns. The uniformity term $P_{\text{uni}}$ reduces overlap, ensuring additional compute translates into new solution modes. Thus, TGR's advantage manifests primarily as higher AUC rather than Pass@1. Furthermore,

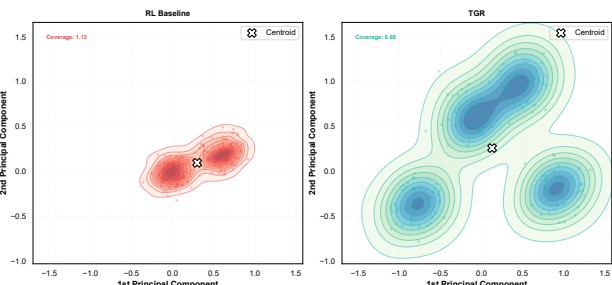 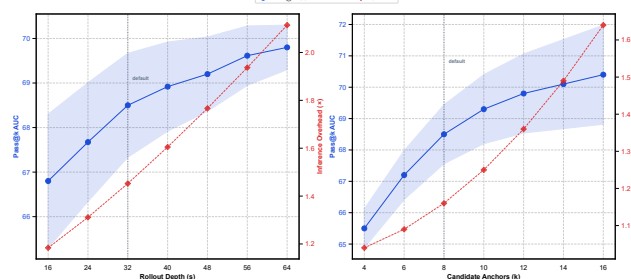

*Figure 4.* **Left: Latent-space mode diversity.** RL-tuned baselines collapse into a unimodal cone, while TGR preserves a well-dispersed distribution, capturing a fuller range of valid reasoning paths. **Right: Hyperparameter robustness.** AUC increases with rollout depth $s$ and beam width $K$, but with diminishing returns.

Table 13 quantifies latent statistics showing that removing $P_{\text{uni}}$ collapses effective candidate-set size $N_{\text{eff}}$.

**Geometry–Robustness Diagnostics.** We find that robustness exhibits a consistent geometric signature, with AUC peaking at an intermediate mean geodesic curvature $\kappa$ that is neither overly rigid nor overly erratic and increasing with the effective candidate-set size proxy $N_{\text{eff}}$. This pattern supports the claim that TGR's advantage stems from maintaining genuinely distinct search paths rather than redundant variations, as illustrated in Figure 13).

## 5. Discussion

### 5.1. Where Geometric Constraints Reside

Recent work interprets Transformer representations as trajectories on a latent manifold, with training objectives shaping its underlying geometry (Aggarwal et al., 2025; Di Sipio et al., 2025; Zhu et al., 2025; Xie et al., 2025). Building on this view, TGR operationalizes geometric structure at inference time through a manifold-informed scoring function over chunk-level candidates, enabling trajectory search without retraining. The key distinction lies in how geometric structure is instantiated. RL internalizes it into model weights, yielding strong single-trajectory priors but concentrating probability mass and limiting controllable exploration (Yue et al., 2025; Srivastava & Aggarwal, 2025). Training-time architectural regularization embeds it into the model parameterization, yet requires modifications to the training pipeline. In contrast, TGR externalizes geometric guidance as an inference-time control layer, offering tunable knobs that directly mediate the trade-off among look-ahead value, coherence, and diversity. This explicit control enhances robustness in long-horizon reasoning by increasing the effective independence of explored trajectories, naturally yielding higher AUC at medium-to-large sampling scales.

### 5.2. Training Stage Determines Inference-Time Controllability

This framing clarifies a key constraint on inference-time controllability, in which score guidance yields larger gains on SFT models than on heavily RL-optimized ones (Figure 5). Once training concentrates the trajectory distribution, degrees of freedom for inference-time search shrink; improvements manifest primarily as stabilizing coverage rather than shifting single-sample accuracy. Conversely, when training does not fully shape long-horizon behavior, explicit inference-time geometry provides a direct mechanism for allocating compute to high-yield branches while maintaining effective parallelism.

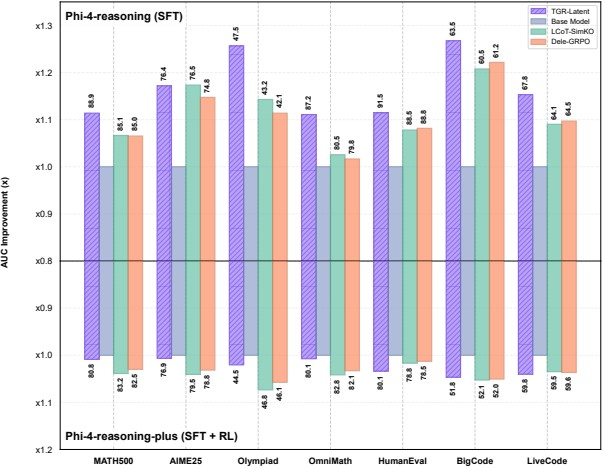

*Figure 5.* **Training stage modulates inference-time controllability.** TGR-Latent yields substantially larger gains on the SFT model (top), while improvement narrows after RL optimization (bottom), suggesting that inference-time search benefits models whose trajectory distribution retains residual flexibility.

### 5.3. Limitations and Future Directions

TGR trades training cost for inference overhead, impacting latency-sensitive serving. Low-overhead rollouts serve as efficient proxies for downstream utility but may be less in-

formative under delayed credit; adaptive budgeting or search at high-uncertainty boundaries offer mitigations. TGR also requires access to hidden states for anchor extraction and residual injection, making open-weight, local, and provider-internal deployments its direct setting; closed-box APIs would require an exposed state-control interface.

Agentic workflows offer significant potential, yet failures frequently arise prior to tool execution due to unstable or ambiguous intent. A low-overhead latent foresight step guided by soft geometric principles could provide a reliable intent anchor prior to tool invocation, offering a potential mechanism to stabilize downstream planning.

## 6. Conclusion

We introduced TGR, a training-free inference-time framework that steers long-horizon reasoning via manifold-informed latent foresight search. By scoring chunk-level anchors with a soft geometric objective combining look-ahead value, coherence, and diversity penalties, TGR matches or exceeds RL-tuned baselines on challenging math and code benchmarks with negligible overhead. Decoupling lightweight boundary rollouts and geometric preferences from model training reveals that robust coverage need not be amortized through training but can instead be induced on the fly via structured latent-space search.

## Impact Statement

This paper studies a training-free, manifold-informed latent foresight search framework for long-context reasoning. If effective, it may improve robustness under fixed or modestly increased compute budgets, benefiting applications such as programming assistance. As with other capability-improving methods, it could also enable misuse (e.g., harmful or misleading content) and its impact depends on deployment context. We recommend pairing such techniques with safeguards and careful evaluation before deployment.

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

# Appendices

## A. Implementation Details of TGR

This appendix supplies implementation details omitted from the main text, focusing on the interface between chunking and latent search. TGR operates on fixed-length token chunks of length $S$, with search performed strictly at chunk boundaries. The base model is frozen, and all auxiliary projections are fixed at initialization.

**Appendix Roadmap.** Appendix B derives complexity and memory properties under the same inference interface; Appendix C defines the evaluation protocol and robustness/efficiency metrics; Appendix D provides additional results; Appendix E reports latent-geometry diagnostics; Appendix F discusses systems/efficiency considerations; Appendix G presents qualitative cases; Appendix H summarizes common failure modes and mitigations.

### A.1. State, Context Windowing, and Chunk Boundaries

At boundary $t - 1$, TGR maintains a pair

$$(c_{t-1},\ z_{t-1}), \tag{11}$$

where $c_{t-1}$ denotes the explicit token context for the next decoding call, and $z_{t-1} \in \mathcal{Z}$ represents the unit-norm state anchor extracted after generating chunk $t - 1$. To limit memory usage, we employ a bounded context window: $c_{t-1}$ always comprises the original query and a recent token suffix, left-truncated to a fixed maximum length. Prior chunks are represented implicitly via the state anchor, avoiding the retention of their full KV cache.

A chunk boundary occurs after generating $S$ tokens, or earlier if a stopping criterion is met. Anchor extraction and search execute only at these boundaries.

### A.2. Anchor Extraction

As defined in Eq. (1), after generating a chunk $\tau_t$ we append a fixed end-of-chunk delimiter token $<\texttt{EOC}>$ and extract the unit-norm state anchor $z_t$ from its top-layer hidden state (equivalently, the last hidden state of $\tau_t \oplus <\texttt{EOC}>$). Using an explicit delimiter provides a consistent extraction position across tasks and stopping behaviors. In our implementation, the projection matrix $W$ is initialized once with i.i.d. Gaussian entries and frozen. When $d_z = d_h$, $W$ can simply be the identity matrix without altering the interface. Thus, semantic information resides in the model-produced $<\texttt{EOC}>$ hidden state rather than in a learned auxiliary projector; $W$ only fixes the coordinate system in which local anchors are compared and perturbed.

### A.3. Low-Rank Conditional Injection

To condition the frozen Transformer on a chosen anchor $a \in \mathcal{Z}$, we employ the low-rank residual stream injection defined in Eq. (8). The injection matrices $(A_\ell, B_\ell)$ are initialized once using Gaussian initialization and then frozen. We set the rank to $r = 8$ by default. This injection applies identically during both candidate evaluation rollouts and full chunk generation. The fixed random interface is used as a stable perturbative channel, not as a separate semantic model. Because candidate rollouts and final chunk generation share the same frozen interface, TGR can compare local trajectory variations consistently; the selected direction is determined by the score in Eq. (3), and Table 3 shows that replacing this choice with a random anchor sharply reduces AUC.

**Efficient Reuse of Prefix Computation.** At each boundary, we rebuild the KV cache for the bounded explicit context $c_{t-1}$ exactly once. Candidate rollouts then reuse this prefix cache and apply anchor injection only during continuation steps, enabling batched parallel evaluation over many candidate anchors with minimal overhead.

### A.4. Candidate Sampling on the Unit Sphere

We generate $K$ candidate anchors by perturbing the previous state anchor $z_{t-1}$ in the tangent space and re-projecting to the sphere, as described in Eq. (2). We utilize a single proposal radius $\sigma$, with default $\sigma = 0.05$. At each boundary, all $K$ candidates are scored in parallel.

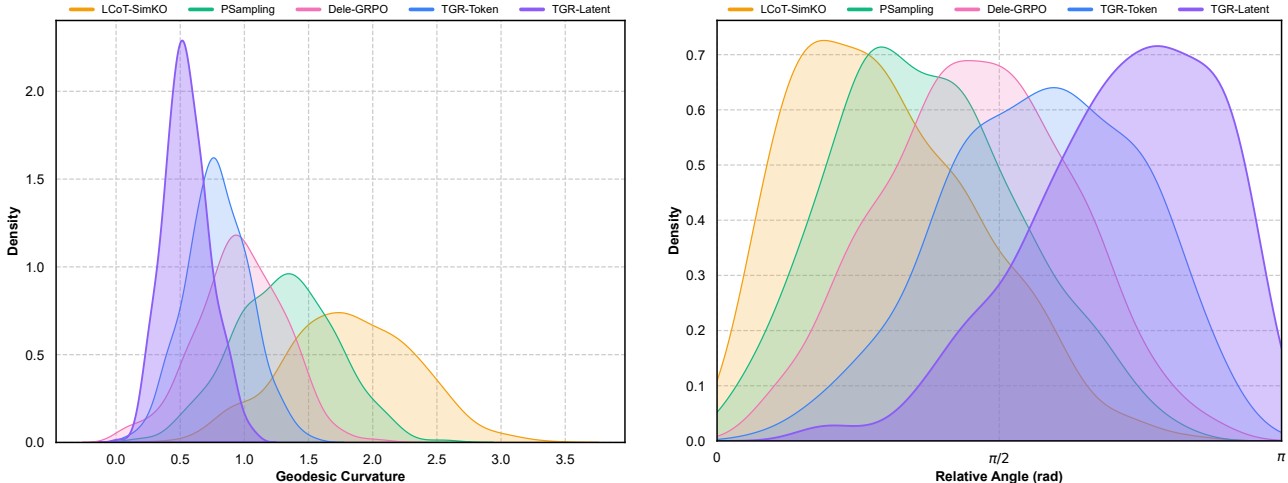

*Figure 6.* **Latent geometry diagnostics on MATH500 with Qwen3-8B. Left: Geodesic curvature $\kappa$.** TGR-Latent concentrates trajectory curvature in a moderate regime, avoiding both rigid collapse and erratic jumps. **Right: Pairwise anchor angle $\theta$.** Angular spread among successive anchors remains broad yet structured, indicating controlled exploration rather than mode collapse or random drift.

## A.5. Score Computation and Parallel Rollouts

Each candidate anchor $\hat{z}$ is evaluated via the manifold-informed scoring function defined in Eq. (3), which combines three components:

- **Foresight Value** $V_{\text{fore}}$ (Eq. 4): We estimate this using a single Monte Carlo rollout per candidate ($N_{\text{MC}} = 1$). Thus, $V_{\text{fore}}$ acts as a single-sample average log-likelihood proxy (Eq. 4 is effectively implemented with $s$ steps). This design minimizes overhead; multi-rollout estimation would scale cost linearly.

- **Bumpiness Penalty** $P_{\text{bum}}$ (Eq. 5): Penalizes high-frequency state variation to ensure coherence. To reduce sensitivity to representation scale, we compute Eq. (5) on $\ell_2$-normalized hidden states.

- **Uniformity Regularizer** $P_{\text{uni}}$ (Eq. 6): A hinge penalty on cosine similarity (with default $\delta = 0.2$) that discourages candidates solely from clustering around the previous anchor $z_{t-1}$.

When rollouts are shallow, $V_{\text{fore}}$ can be noisy for near-tied candidates; a variance-reduction option is to re-evaluate only the top few candidates (or temporarily increase $N_{\text{MC}}$) at a small marginal cost (see Appendix C.9).

## A.6. Latent Geometry Diagnostics

**Geometry Statistics: Intermediate Curvature with Controlled Angular Spread.** Figure 6 summarizes trajectory-level geometry. TGR-Latent concentrates probability mass toward a moderate-curvature regime while maintaining a broad, structured angular spread, consistent with stabilized long-horizon steering that avoids both rigid collapse and unconstrained drift.

**Scoring Diagnostics: Selection Is Driven by Look-Ahead Value, Regularized by Geometry.** Figure 7 (left) shows that the score components play complementary roles: the look-ahead term provides the primary separation signal, while bumpiness and uniformity penalties stabilize candidates against pathological turns and collapse. Figure 7 (right) further confirms that selected anchors exhibit consistently higher $V_{\text{fore}}$ than rejected ones, indicating that candidate choice is driven by lightweight look-ahead rather than local likelihood spikes.

## A.7. Full Inference Procedure

Algorithm 2 provides a complete, self-contained version of TGR inference, including candidate sampling, scoring, and chunk generation. The main paper uses a compressed form.

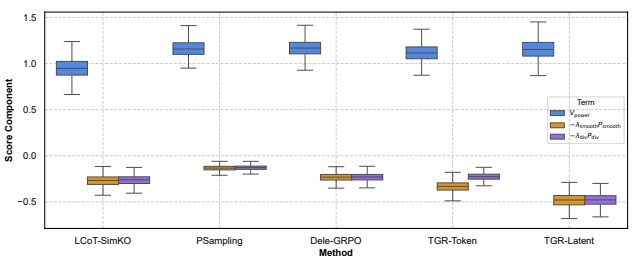
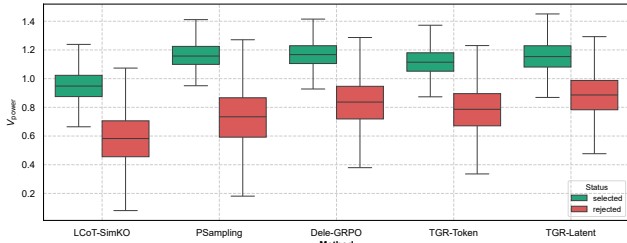

*Figure 7.* **Anchor scoring and selection mechanisms in TGR. Left: Score component decomposition across methods.** Box plots show the distribution of each score term ($V_{\text{fore}}$, $-\lambda_b P_{\text{bum}}$, $-\lambda_u P_{\text{uni}}$). The look-ahead value $V_{\text{fore}}$ provides the dominant separation signal, while the bumpiness and uniformity penalties contribute stabilizing regularization. **Right: Foresight value distinguishes selected from rejected anchors.** Across all methods, selected anchors exhibit higher $V_{\text{fore}}$ than rejected ones, confirming that TGR's candidate selection is driven by lightweight look-ahead utility rather than local likelihood peaks.

---

**Algorithm 2** Full TGR Inference Process

---

1: **Input:** prompt/question $Q$; chunk length $S$; max chunks $L$; candidate anchors $K$; rollout length $s$.
2: **Fixed components:** projection $W$ (Eq. 1); low-rank injectors $\{A_\ell, B_\ell\}$ (Eq. 8); hyperparameters $\sigma, \lambda_b, \lambda_u, \delta$.
3: Initialize explicit context $c_0 \leftarrow Q$ (apply windowing); run the base model once to obtain $z_0$ via Eq. 1.
4: **for** $t = 1$ **to** $L$ **do**
5:      Build (or rebuild) prefix KV cache for $c_{t-1}$ once.
6:      **for** $j = 1$ **to** $K$ **do**
7:          Sample candidate anchor $\hat{z}^{(j)}$ around $z_{t-1}$ with radius $\sigma$ by Eq. 2.
8:          Roll out $\hat{\tau}^{(j)}$ for $s$ steps from $c_{t-1}$, conditioning on $\hat{z}^{(j)}$ via Eq. 8.
9:          Compute $V_{\text{fore}}$ (Eq. 4), $P_{\text{bum}}$ (Eq. 5), $P_{\text{uni}}(\hat{z}^{(j)}; z_{t-1})$ (Eq. 6).
10:          $\mathcal{S}^{(j)} \leftarrow V_{\text{fore}} - \lambda_b P_{\text{bum}} - \lambda_u P_{\text{uni}}$.
11:      **end for**
12:      Select $j^\star \leftarrow \arg\max_j \mathcal{S}^{(j)}$.
13:      $\hat{z}_t \leftarrow \hat{z}^{(j^\star)}$.
14:      Generate a full chunk $\tau_t$ of length $S$ from context $c_{t-1}$, conditioned on $\hat{z}_t$ via Eq. 8.
15:      Extract new state anchor $z_t$ from $\tau_t$ via Eq. 1.
16:      Update explicit context $c_t \leftarrow \text{Window}(Q \oplus \tau_t)$.
17:      Optional: stop early if a termination condition is met.
18: **end for**
19: **Output:** completion trajectory $\tau_{1:T}$.

---

# B. Additional Theoretical Details: Complexity, Memory, and Hard Geometry

This appendix expands upon § 3 with explicit derivations under the exact inference interface used by TGR.

## B.1. Runtime Decomposition in Forward-Token Units

We assess computational cost via forward-token units: the total token positions processed by the forward pass. This captures candidate rollout costs, even when tokens are discarded.

**Complexity.** Under the standard quadratic attention cost model, generating one chunk of length $S$ costs $O(S^2)$, and scoring $K$ candidate anchors via rollouts of depth $s \ll S$ costs $O(Ks^2)$ per chunk step. Thus, the overall time complexity over $L$ chunk steps is

$$\text{Time} = O(L(S^2 + Ks^2)) = O(LS^2), \tag{12}$$

when $s \ll S$ and $K$ is treated as a constant. We use forward-token accounting below for a more implementation-faithful decomposition, and defer KV-memory bounds to Appendix B.2.

Let $L$ be the maximum number of chunks, $S$ the chunk length, and $K$ the number of candidate anchors scored per boundary. Let $s \ll S$ denote the rollout depth. Let $c_{\text{tok}}$ be the amortized per-token forward cost of the frozen base model (attention + MLP).

**Main-Path Chunk Generation.** At each chunk boundary, TGR generates a single full chunk of length $S$. Hence, the forward-token cost of main-path generation is

$$T_{\text{main}} = \mathcal{O}(LS \cdot c_{\text{tok}}). \tag{13}$$

**Candidate Rollouts.** At each boundary, we evaluate $K$ candidate anchors via a lightweight rollout of length $s$. This contributes

$$T_{\text{roll}} = \mathcal{O}(LKs \cdot c_{\text{tok}}). \tag{14}$$

The geometric penalties are computed from hidden states or latent vectors and are negligible compared to forward passes.

**Total Runtime and Multiplier.** Combining Eq. 13 and Eq. 14, the total forward-token runtime is

$$T_{\text{TGR}} = T_{\text{main}} + T_{\text{roll}} = \mathcal{O}\Big(L(S + Ks) \cdot c_{\text{tok}}\Big). \tag{15}$$

Relative to the main-path generation cost $T_{\text{main}}$, the overhead multiplier is approximately

$$\frac{T_{\text{TGR}}}{T_{\text{main}}} \approx 1 + \frac{Ks}{S}. \tag{16}$$

Equation 16 highlights the central design trade-off: low-depth rollouts ($s \ll S$) and modest candidate budget ($K$) keep overhead small while preserving look-ahead.

## B.2. Memory: Chunk-Wise KV Reset and the $O(S)$ Bound

The dominant inference-time memory is the KV cache. For a Transformer with $n_\ell$ layers and width $d_h$, the cache footprint scales linearly with cached tokens $n_{\text{ctx}}$:

$$M_{\text{KV}} = \Theta(n_\ell\, n_{\text{ctx}}\, d_h), \tag{17}$$

up to constants from heads and precision.

**Unbounded Caching.** Without resets, after $t$ chunks the context length grows as $n_{\text{ctx}} = \Theta(tS)$, leading to peak memory $\Theta(n_\ell\, tS\, d_h)$, which becomes prohibitive for exploration.

**TGR: Bounded Explicit Context.** TGR resets the KV cache at chunk boundaries and rebuilds it from a bounded explicit context that retains the question/prompt and the most recent chunk:

$$c_{t-1} \triangleq Q \oplus \tau_{t-1}. \tag{18}$$

Earlier history is not kept in the KV cache and is instead carried implicitly through the injected latent anchor (Appendix A). Therefore,

$$n_{\text{ctx}} = \Theta(|Q| + S), \tag{19}$$

and the peak KV cache footprint satisfies

$$M_{\text{KV}} = \Theta\big(n_\ell(|Q| + S)d_h\big) = \mathcal{O}(S), \tag{20}$$

treating $|Q|$ as fixed for a given query. Candidate rollouts reuse the same bounded-context cache and do not change this asymptotic bound.

**Relation to KV-Cache Compression/Retention.** Systems techniques including KV-cache compression, merging, or selective retention primarily reduce constants or the effective cached length under a given retention policy. TGR is complementary: it changes the inference interface by enforcing a bounded explicit context (Eq. 18 to Eq. 19), yielding an algorithmic $O(S)$ KV bound independent of the full reasoning trace length. We do not include an empirical comparison to these systems methods in this paper.

Table 5 summarizes this KV-memory scaling comparison.

*Table 5.* **KV-memory scaling comparison.** TGR's chunk-wise reset bounds cache size to $O(S)$ regardless of horizon length, whereas unbounded caching grows linearly with total generated tokens. Here $t$ is the chunk index, $S$ the chunk length, and $|Q|$ the prompt length.

| Method/interface | Active cached tokens $n_{\text{ctx}}$ | KV memory scaling |
|---|---|---|
| Unbounded KV caching | $\Theta(|Q| + tS)$ | $\Theta(n_\ell(|Q| + tS)d_h)$ |
| Chunk-wise KV reset | $\Theta(|Q| + S)$ | $\Theta(n_\ell(|Q| + S)d_h) = \mathcal{O}(S)$ |
| TGR (this paper) | $\Theta(|Q| + S)$ | $\Theta(n_\ell(|Q| + S)d_h) = \mathcal{O}(S)$ |

## B.3. Vanishing Acceptance in Hard Geometric Constraints

This section formalizes the acceptance rate $\alpha$ for hard geometric constraints, i.e., Manifold-Constrained feasibility filtering, and explains why strict feasibility filtering becomes impractical at scale.

### B.3.1. ACCEPTANCE RATE AND GEOMETRY

As formally defined in § 3.5 (Eq. 9), the acceptance rate $\alpha$ represents the probability that a random proposal satisfies the hard geometric constraint. The associated compute multiplier for rejection sampling is approximately $1/\alpha$. In this section, we provide a geometric intuition for why this rate collapses exponentially, and report empirical measurements.

### B.3.2. WHY $\alpha$ COLLAPSES IN HIGH DIMENSIONS (HEURISTIC INTUITION)

We provide a geometric intuition using concentration of measure on the unit sphere. Many hard constraints require candidates to lie inside a small admissible region (e.g., a narrow spherical cap). Consider a rule enforcing near-alignment with the parent anchor:

$$\mathcal{G}(\hat{z}; z_{t-1}) = 1 \quad \Leftrightarrow \quad \cos(\hat{z}, z_{t-1}) \geq \rho, \tag{21}$$

for a fixed $\rho \in (0, 1)$. For a random unit vector $\hat{z}$, $\cos(\hat{z}, z_{t-1})$ concentrates near 0 with variance on the order of $1/d$ in $d$ dimensions. A standard tail heuristic yields

$$\alpha = \Pr\left[\cos(\hat{z}, z_{t-1}) \geq \rho\right] \approx \exp\left(-\tfrac{1}{2}d\rho^2\right), \tag{22}$$

demonstrating that the admissible measure shrinks rapidly as effective dimension grows. While practical constraints, for example, geodesic-following feasibility, are more structured than Eq. 21, they similarly correspond to selecting a vanishing fraction of candidate directions as dimension increases.

### B.3.3. EMPIRICAL FEASIBILITY RATES

We report feasibility acceptance rates in Table 6 for a fixed hard geometric constraint (geodesic-following feasibility) across model scales, measured as the fraction of sampled candidates that satisfy the constraint. The implied multiplier $1/\alpha$ highlights the practical infeasibility of strict rejection at larger scales.

*Table 6.* **Hard geometric constraints become infeasible at scale.** Feasibility acceptance rates $\alpha$ for a curvature-bounded geodesic-following constraint across Qwen3 scales. At 14B, fewer than 1.5% of proposals pass the hard filter, requiring $> 66\times$ compute inflation. TGR avoids this blow-up via soft regularization.

| Model scale | Acceptance $\alpha$ | Compute mult. $(1/\alpha)$ |
|---|---|---|
| 1.7B | 28.5% | $\sim 3.5\times$ |
| 8B | 9.8% | $\sim 10.2\times$ |
| 14B | $< 1.5\%$ | $> 66\times$ |

**Soft Geometry Avoids Rejection Blow-Up.** TGR replaces hard geometric feasibility gating with soft geometric penalties, turning geometric priors into continuous costs (a manifold-informed regularization) rather than strict accept/reject feasibility. Instead of forcing candidates to fall inside an exponentially small feasible set, TGR assigns costs to curvature and redundancy, preserving controllable exploration while maintaining bounded memory via chunk-wise KV resets.

## B.4. Architecture Versus Algorithm

Comparing GRPO and SimKO under structured state-transition architectures suggests that architectural inductive bias can dominate the optimization objective. Delethink-style state isolation and discrete-state interfaces stabilize single-trajectory success by reducing error propagation, potentially at the cost of constrained exploration and reduced effective diversity. In contrast, TGR retains a continuous, manipulable latent control variable and enforces diversity via a soft repulsion term, which preserves parallel exploration without relying on training-time objectives. This perspective helps explain why TGR can match or exceed computationally intensive RL baselines in robust coverage under a matched inference budget.

# C. Experimental Setup Details and Metrics

This appendix details the experimental protocol supporting § 4, including unified compute accounting and Pass@$k$/AUC definitions. Unless stated otherwise, we evaluate all methods using the same sampling grid and decoding configuration.

## C.1. Unified Inference Budget and Decoding Configuration

All methods operate under a unified inference budget with a fixed maximum horizon ($L = 24$ chunks, maximum chunk length $S = 512$) and temperature $T = 0.6$. At each chunk boundary, TGR-Latent evaluates $K = 8$ candidate anchors using lightweight rollouts of length $s = 32$ for math tasks and $s = 64$ for code tasks, then executes the highest-scoring one.

## C.2. Benchmarks and Evaluation Suites

We evaluate on mathematical reasoning and code generation benchmarks: MATH500 (Hendrycks et al., 2021), AIME2025 (MAA, 2025), OlympiadBench (He et al., 2024), OmniMath (Gao et al., 2024), HumanEval (Chen, 2021), BigCodeBench (Zhuo et al., 2024), and LiveCodeBench (Jain et al., 2024). For each benchmark, we follow its standard evaluation protocol and official scoring scripts when available. Appendix D.3 further reports pilot open-ended evaluations on IFEval (Zhou et al., 2023) and the Literature & Arts subset of WritingBench (Wu et al., 2025a).

## C.3. Sampling Grid for Pass@$k$ and AUC

We evaluate robustness on a full log-spaced sampling grid

$$\mathcal{K} = \{2^0, 2^1, \ldots, 2^7\} = \{1, 2, 4, 8, 16, 32, 64, 128\}. \tag{23}$$

Intermediate points (Eq. 23) are obtained by running inference at the corresponding budget, not interpolation. Main tables report key points ($k \in \{1, 32, 128\}$); AUC and Pass@$k$ curves use the full grid. For clarity, $k$ denotes the number of final trajectories generated per problem for Pass@$k$, while $K$ denotes the within-step candidate-anchor pool size used internally by TGR at each chunk boundary. For TGR, producing $k$ trajectories means running the full TGR procedure end-to-end $k$ times with independent decoding randomness; internal candidate rollouts used for scoring are accounted for in total forward tokens (Sec. C.6) but do not count as additional final trajectories.

## C.4. Pass@$k$ and the Unbiased Estimator

For code generation tasks, we use the unbiased Pass@$k$ estimator from Chen et al. (Chen, 2021). Let $n$ be the number of generated samples and $c$ be the number of correct samples under the benchmark verifier. The unbiased estimate is

$$\widehat{\text{Pass}}@k = \begin{cases} 1 - \dfrac{\binom{n-c}{k}}{\binom{n}{k}}, & n - c \geq k, \\ 1, & n - c < k. \end{cases} \tag{24}$$

For mathematical reasoning benchmarks, Pass@$k$ is calculated via the dataset-specific correctness criterion, such as exact answer match, under the same sampling grid $\mathcal{K}$.

## C.5. AUC over the Log-Scaled Sampling Budget

To quantify robustness across the sampling budget, we report the normalized Area Under the Curve (AUC) on a log-scale as formally defined in Eq. 10 (§ 4). The metric is computed via trapezoidal integration over the $\log_2 k$ grid, normalized

*Table 7.* **Default hyperparameters and inference settings.** All methods share the global decoding configuration. TGR-specific parameters control latent search depth and geometric regularization. Baselines are evaluated under matched end-to-end token budgets.

| Category | Setting |
|---|---|
| *Global decoding (all methods)* | |
| Temperature | 0.6 |
| Sampling grid $\mathcal{K}$ | $\{2^0, 2^1, \ldots, 2^7\}$ |
| Chunks / chunk length | $L = 24, S = 512$ |
| Token accounting | Total forward tokens (incl. rollouts and KV rebuild) |
| *TGR-Latent (search)* | |
| Candidate anchors per boundary | $K = 8$ |
| Rollout depth | $s = 32$ (math), $s = 64$ (code) |
| MC rollouts per candidate | $N_{\mathrm{MC}} = 1$ |
| Proposal radius | $\sigma = 0.05$ |
| Repulsion threshold | $\delta = 0.2$ |
| Score weights | $\lambda_b = 0.5, \; \lambda_u = 0.5$ |
| Context management | Chunk-wise KV reset; $c_{t-1} = Q \oplus \tau_{t-1}$ |
| Injection rank | $r = 8$ (fixed random init; not trained) |
| *Baselines (key settings)* | |
| PSampling | $T = 0.6$; budget matched on total forward tokens |
| LCoT / Delethink | Standard sampling; budget matched on total forward tokens |
| TGR-Token | Token-space ablation (branching; $K = 8$) |

to the range $[0, 100]$. This approach ensures that performance improvements at higher sampling budgets are weighted proportionally to the exponential increase in compute.

## C.6. Total Forward Tokens

We report Avg. Tokens as total forward tokens consumed per problem: (i) full chunk generation, (ii) candidate scoring rollouts, and (iii) KV cache rebuilding. This convention prevents hidden search cost, enabling faithful efficiency comparison.

## C.7. Reproducibility Notes

All methods are evaluated under the same sampling grid $\mathcal{K}$ and decoding settings. When a method introduces additional internal steps, for example, candidate scoring rollouts, their compute is fully reflected in the reported Avg. Tokens as defined in Sec. C.6. Our comparisons are restricted to methods that can be evaluated under this unified token-accounting protocol without introducing additional external models or task-specific scoring infrastructure; we additionally include TGR-Token to isolate token-space structured control under the same chunked interface. Self-consistency-style aggregation is orthogonal to our coverage-focused Pass@$k$/AUC evaluation, and can be applied as a post-processing step on top of any sampling method. For repeated-run statistics, we use three independent runs with seeds 42, 1234, and 3407. In each run, AUC is computed over the full grid $\mathcal{K}$ in Eq. 23.

## C.8. Hyperparameters

Table 7 summarizes the default hyperparameters and inference settings used throughout our experiments. Unless otherwise noted, all methods share the same global decoding configuration, and all comparisons are conducted under end-to-end compute accounting.

**Conventions.** We evaluate on the full log-spaced sampling grid $\mathcal{K}$ (Appendix C.3) and report Avg. Tokens using end-to-end forward-token accounting (Appendix C.6).

**TGR Variants and Defaults.** TGR-Latent performs latent-space candidate search with low-rank injection, while TGR-Token is its token-space analogue under the same chunked interface and compute accounting. Default values for $K, s, \sigma, \lambda_b, \lambda_u, \delta, r$ are listed in Table 7.

*Table 8.* **Sensitivity to latent-interface parameters on Qwen3-8B.** TGR maintains robust AUC and low overhead across a wide range of settings, with performance degrading only under extreme configurations.

| Setting | Avg. AUC | Overhead | Trend |
|---|---|---|---|
| Default ($d_z = d_h, r = 8, \eta = 1, N_{MC} = 1$) | 53.2 | $\sim 1.11\times$ | — |
| *Injection rank $r$* | | | |
|     $r = 4$ | 52.8 | $\sim 1.11\times$ | $\downarrow$ control |
|     $r = 16$ | 53.1 | $\sim 1.11\times$ | $\leftrightarrow$ |
|     $r = 32$ | 52.2 | $\sim 1.11\times$ | $\downarrow$ stability |
| *Anchor dim. $d_z$* | | | |
|     $d_z = d_h/2$ | 52.7 | $\sim 1.11\times$ | $\downarrow$ capacity |
|     $d_z = 256$ | 51.8 | $\sim 1.11\times$ | $\downarrow\downarrow$ bottleneck |
| *Injection scale $\eta$* | | | |
|     $\eta = 0.5$ | 52.9 | $\sim 1.11\times$ | $\downarrow$ foresight |
|     $\eta = 2.0$ | 51.9 | $\sim 1.11\times$ | $\downarrow$ stability |
| *Rollout strategy* | | | |
|     $N_{MC} = 2$ (fixed $K$) | 53.3 | $\sim 1.17\times$ | $\uparrow$ signal / $\uparrow$ cost |
|     Top-2 re-eval ($N_{MC} = 1$) | 53.5 | $\sim 1.14\times$ | $\downarrow$ variance |

### C.9. Sensitivity to the Latent Control Interface

TGR employs fixed random matrices to enable training-free latent control. We summarize a sensitivity sweep over (i) anchor dimension $d_z$, (ii) injection rank $r$, (iii) injection scale $\eta$, and (iv) Monte Carlo rollouts $N_{MC}$ in Table 8.

## D. Additional Results and Diagnostics

### D.1. Scaling Across Model Sizes

Tables 9 and 10 report results on all Qwen3 scales from 1.7B to 14B under a unified budget. TGR-Latent consistently improves both math and code, suggesting that latent search remains effective across model scales.

*Table 9.* **Mathematical reasoning across Qwen3 scales under unified budget.** TGR-Latent achieves the best or near-best AUC and high-$k$ accuracy at every scale while consuming fewer total forward tokens than computationally intensive RL baselines, confirming that inference-time latent search generalizes across model capacities.

| Method | AIME25 | | | | OlympiadBench | | | | OmniMath | | | | Tokens |
|---|---|---|---|---|---|---|---|---|---|---|---|---|---|
| | @1 | @32 | @128 | AUC | @1 | @32 | @128 | AUC | @1 | @32 | @128 | AUC | Avg. ($\times 10^3$) |
| | | | | | | | QWEN3-1.7B | | | | | | |
| Base Model | 3.1 | 7.5 | 8.2 | 5.8 | 5.4 | 11.1 | 12.1 | 9.0 | 8.7 | 17.5 | 19.3 | 14.3 | 1.5 |
| PSampling | 4.5 | 8.6 | 9.4 | 7.2 | 7.2 | 12.6 | 13.8 | 10.8 | 15.1 | 20.8 | 22.1 | 18.2 | 3.5 |
| LCoT-GRPO | 6.0 | 7.8 | 8.3 | 7.2 | 9.1 | 11.8 | 12.3 | 10.8 | 14.2 | 18.5 | 19.9 | 17.3 | 4.8 |
| LCoT-SimKO | 6.5 | 9.5 | 10.2 | 8.5 | 9.8 | 13.5 | 14.5 | 12.4 | 14.9 | 20.1 | 21.5 | 18.5 | 4.9 |
| Dele-GRPO | 6.5 | 11.2 | 12.0 | 9.5 | 9.6 | 15.1 | 16.2 | 13.2 | 14.8 | 23.0 | 24.8 | 20.2 | 4.6 |
| Dele-SimKO | **7.0** | **12.1** | **13.0** | **10.2** | **10.3** | **16.0** | **17.2** | **14.1** | **15.8** | **24.1** | **26.0** | **21.3** | 4.7 |
| TGR-Token | 5.1 | 9.8 | 10.8 | 8.2 | 8.0 | 14.0 | 15.2 | 12.0 | 12.9 | 22.3 | 24.3 | 19.2 | 4.3 |
| TGR-Latent | 5.8 | 11.5 | 12.5 | 9.4 | 9.0 | 15.9 | 17.1 | 13.7 | 13.8 | 23.5 | 25.9 | 20.5 | 3.9 |
| | | | | | | | QWEN3-4B | | | | | | |
| Base Model | 8.4 | 15.0 | 16.3 | 12.7 | 12.8 | 20.5 | 22.4 | 18.0 | 19.3 | 32.0 | 34.7 | 27.7 | 2.2 |
| PSampling | 11.2 | 17.5 | 18.9 | 15.4 | 16.1 | 23.5 | 25.3 | 21.2 | 24.7 | 36.0 | 38.2 | 32.1 | 5.5 |
| LCoT-GRPO | 13.5 | 15.8 | 16.5 | 15.2 | 19.0 | 21.8 | 22.8 | 21.0 | 27.5 | 34.0 | 35.5 | 31.9 | 7.8 |
| LCoT-SimKO | 14.1 | 18.5 | 19.8 | 17.2 | 20.1 | 25.0 | 26.5 | 23.5 | 28.8 | 37.1 | 39.0 | 34.1 | 7.9 |
| Dele-GRPO | 14.1 | 21.8 | 23.5 | 19.3 | 20.0 | 28.1 | 30.0 | 25.5 | 28.6 | 40.1 | 42.4 | 36.2 | 7.2 |
| Dele-SimKO | **14.8** | 23.1 | 25.0 | **20.4** | 20.8 | 29.8 | 32.0 | **27.0** | **29.7** | 41.8 | 44.2 | **37.8** | 7.4 |
| TGR-Token | 12.5 | 19.8 | 21.4 | 17.4 | 17.8 | 26.2 | 28.0 | 23.4 | 26.9 | 39.1 | 41.5 | 34.9 | 7.0 |
| TGR-Latent | 14.0 | **23.5** | **25.5** | **20.4** | 19.8 | **30.1** | **32.8** | 26.9 | 28.8 | **42.1** | **44.8** | 37.7 | 6.5 |
| | | | | | | | QWEN3-8B | | | | | | |
| Base Model | 18.7 | 26.5 | 28.4 | 24.0 | 22.1 | 30.2 | 32.6 | 27.9 | 31.4 | 45.8 | 48.9 | 41.2 | 2.6 |
| PSampling | 23.8 | 30.1 | 32.1 | 28.4 | 27.4 | 34.5 | 36.8 | 32.6 | 38.7 | 49.8 | 52.3 | 46.2 | 6.0 |
| LCoT-GRPO | 26.5 | 27.8 | 28.5 | 27.5 | 30.1 | 31.8 | 32.5 | 31.5 | 41.8 | 48.0 | 49.8 | 46.2 | 8.5 |
| LCoT-SimKO | 27.5 | 31.5 | 33.0 | 30.5 | 31.8 | 36.0 | 38.0 | 35.1 | 43.5 | 51.5 | 54.0 | 49.1 | 8.6 |
| Dele-GRPO | 27.4 | 35.5 | 37.8 | 33.2 | 31.5 | 39.0 | 41.8 | 37.2 | 43.1 | 54.2 | 57.1 | 50.9 | 8.4 |
| Dele-SimKO | **28.4** | 37.0 | 39.5 | **34.8** | 32.5 | 40.8 | 43.8 | 38.8 | **44.6** | 56.1 | 59.2 | **52.8** | 8.5 |
| TGR-Token | 25.4 | 33.2 | 35.6 | 31.1 | 29.1 | 37.0 | 39.5 | 34.8 | 41.2 | 53.0 | 55.8 | 49.3 | 7.5 |
| TGR-Latent | 27.8 | **37.4** | **40.1** | **34.8** | 32.8 | **41.8** | **44.9** | **39.5** | 43.8 | **56.8** | **60.1** | **52.8** | 7.0 |
| | | | | | | | QWEN3-14B | | | | | | |
| Base Model | 32.1 | 40.5 | 42.8 | 38.1 | 34.2 | 42.5 | 45.1 | 40.2 | 48.3 | 59.8 | 62.7 | 56.3 | 2.9 |
| PSampling | 38.9 | 44.2 | 46.2 | 42.9 | 40.1 | 46.1 | 48.3 | 44.6 | 55.7 | 63.5 | 65.8 | 61.3 | 6.5 |
| LCoT-GRPO | 43.2 | 45.0 | 45.8 | 44.6 | 44.8 | 46.8 | 47.8 | 46.3 | 58.5 | 62.5 | 63.8 | 61.3 | 9.2 |
| LCoT-SimKO | 44.5 | 48.5 | 50.1 | 47.6 | 46.5 | 50.5 | 52.5 | 49.7 | 60.1 | 65.5 | 67.5 | 64.0 | 9.3 |
| Dele-GRPO | 44.5 | 50.5 | 53.2 | 49.3 | 46.1 | 52.0 | 54.9 | 51.0 | 59.9 | 67.9 | 70.8 | 65.9 | 8.9 |
| Dele-SimKO | **45.6** | 52.1 | 55.0 | 50.9 | 47.1 | 53.8 | 56.9 | 52.6 | **61.2** | 69.5 | 72.3 | 67.5 | 9.1 |
| TGR-Token | 41.3 | 47.3 | 49.5 | 45.8 | 42.8 | 49.0 | 51.2 | 47.5 | 58.4 | 66.5 | 68.9 | 64.3 | 7.6 |
| TGR-Latent | 44.8 | **52.6** | **55.5** | 50.8 | **47.5** | **54.8** | **58.0** | **53.4** | 60.5 | **70.5** | **73.8** | **67.8** | 7.3 |

*Table 10.* **Code generation across Qwen3 scales under unified budget.** With end-to-end forward-token accounting, TGR-Latent achieves the top or near-top AUC and high-$k$ accuracy on all three benchmarks, demonstrating favorable speed–quality trade-offs for practical code synthesis.

| Method | HumanEval | | | | BigCodeBench | | | | LiveCodeBench | | | | Tokens |
|---|---|---|---|---|---|---|---|---|---|---|---|---|---|
| | @1 | @32 | @128 | AUC | @1 | @32 | @128 | AUC | @1 | @32 | @128 | AUC | Avg. ($\times 10^3$) |
| | | | | | | QWEN3-1.7B | | | | | | | |
| Base Model | 18.3 | 38.5 | 41.5 | 31.8 | 12.1 | 26.0 | 28.7 | 21.6 | 8.4 | 17.5 | 19.3 | 14.6 | 1.2 |
| PSampling | 24.7 | 46.5 | 49.8 | 39.2 | 16.8 | 31.2 | 34.2 | 26.9 | 12.1 | 22.6 | 24.6 | 19.4 | 2.8 |
| LCoT-GRPO | 26.8 | 49.2 | 52.0 | 42.1 | 18.8 | 33.8 | 36.1 | 28.8 | 13.5 | 24.2 | 26.2 | 21.0 | 4.9 |
| LCoT-SimKO | _27.9_ | 51.5 | 54.2 | _44.5_ | 19.5 | 35.5 | 38.0 | 30.2 | 14.1 | 26.0 | 28.1 | 22.4 | 5.0 |
| Dele-GRPO | _27.9_ | 52.5 | 55.8 | 44.1 | **20.1** | 36.0 | 38.8 | _30.7_ | _14.3_ | 26.6 | 28.9 | 22.8 | 4.9 |
| Dele-SimKO | **28.4** | _53.2_ | _56.5_ | **45.2** | _19.8_ | **36.8** | **39.5** | **31.1** | **14.6** | _27.2_ | _29.8_ | **23.4** | 4.8 |
| TGR-Token | 26.9 | 50.8 | 54.5 | 42.8 | 18.5 | 35.1 | 37.8 | 29.3 | 13.7 | 25.5 | 27.9 | 21.7 | 3.6 |
| TGR-Latent | 27.1 | **53.8** | **57.2** | 44.4 | 18.9 | _36.2_ | _38.9_ | 30.4 | 14.2 | **27.9** | **30.4** | _23.2_ | 3.5 |
| | | | | | | QWEN3-4B | | | | | | | |
| Base Model | 32.1 | 55.1 | 58.7 | 46.8 | 22.3 | 36.2 | 38.9 | 31.5 | 16.7 | 26.5 | 28.4 | 23.1 | 1.4 |
| PSampling | 39.4 | 62.5 | 65.8 | 54.8 | 28.7 | 41.5 | 44.5 | 37.4 | 21.8 | 31.7 | 33.7 | 28.3 | 3.2 |
| LCoT-GRPO | 41.5 | 64.8 | 67.8 | 57.2 | 31.0 | 44.5 | 47.0 | 39.8 | 23.8 | 33.5 | 35.8 | 30.1 | 5.6 |
| LCoT-SimKO | 42.8 | 67.2 | 70.5 | 59.5 | 31.9 | 46.8 | 49.2 | 41.7 | _24.9_ | 35.8 | 38.0 | 32.1 | 5.7 |
| Dele-GRPO | 42.6 | 68.0 | 71.5 | 59.4 | 31.8 | 47.5 | 50.1 | 42.0 | _24.9_ | 36.2 | 38.4 | 32.3 | 5.6 |
| Dele-SimKO | **43.2** | **69.5** | _72.5_ | **60.8** | **32.5** | **48.3** | _51.0_ | **42.8** | **25.4** | _37.0_ | _39.1_ | _33.0_ | 5.4 |
| TGR-Token | 41.8 | 66.5 | 70.0 | 58.2 | 30.9 | 45.8 | 48.2 | 40.5 | 23.9 | 34.5 | 36.8 | 30.9 | 4.0 |
| TGR-Latent | 42.5 | _69.2_ | **73.0** | _60.5_ | _32.0_ | _48.1_ | **51.3** | _42.7_ | 24.8 | **37.5** | **39.7** | **33.1** | 4.1 |
| | | | | | | QWEN3-8B | | | | | | | |
| Base Model | 45.7 | 67.8 | 71.2 | 60.3 | 34.1 | 48.5 | 51.3 | 43.8 | 26.8 | 36.8 | 38.9 | 33.5 | 1.6 |
| PSampling | 52.8 | 74.5 | 77.8 | 67.2 | 40.8 | 54.2 | 57.2 | 49.9 | 32.4 | 42.1 | 44.1 | 38.8 | 3.6 |
| LCoT-GRPO | 55.0 | 76.8 | 79.5 | 69.5 | 43.5 | 57.5 | 60.2 | 52.8 | 35.5 | 44.5 | 46.8 | 41.5 | 6.0 |
| LCoT-SimKO | 56.2 | 79.2 | 82.0 | 71.8 | 44.8 | 60.1 | 62.8 | 55.0 | 36.5 | 46.5 | 48.8 | 43.2 | 6.0 |
| Dele-GRPO | _56.4_ | 79.8 | 82.5 | 71.2 | 44.8 | 60.5 | 63.3 | 55.1 | 36.4 | 46.8 | 49.1 | 43.4 | 5.9 |
| Dele-SimKO | **57.1** | _80.5_ | _83.5_ | _73.2_ | _45.5_ | _61.3_ | _64.1_ | _56.0_ | **37.2** | _47.5_ | _49.8_ | _44.1_ | 5.8 |
| TGR-Token | 55.3 | 78.5 | 81.8 | 70.8 | 43.5 | 58.8 | 61.4 | 53.6 | 35.1 | 45.5 | 47.8 | 42.0 | 4.4 |
| TGR-Latent | 56.1 | **81.1** | **84.2** | **73.5** | **45.6** | **62.1** | **64.8** | **56.3** | _36.8_ | **48.1** | **50.2** | **44.4** | 4.6 |
| | | | | | | QWEN3-14B | | | | | | | |
| Base Model | 58.9 | 77.0 | 79.8 | 70.4 | 46.2 | 59.5 | 62.4 | 55.1 | 37.1 | 46.2 | 48.3 | 43.1 | 1.8 |
| PSampling | 65.4 | 83.2 | 85.5 | 76.8 | 52.8 | 65.9 | 67.9 | 61.2 | 43.2 | 51.7 | 53.7 | 48.1 | 3.8 |
| LCoT-GRPO | 67.5 | 84.5 | 86.8 | 78.9 | 55.5 | 68.0 | 70.5 | 63.8 | 46.5 | 54.5 | 56.8 | 51.5 | 6.4 |
| LCoT-SimKO | 68.8 | 86.2 | 88.5 | 80.5 | 56.8 | 70.1 | 72.8 | 66.1 | 47.5 | 56.8 | 59.1 | 53.8 | 6.4 |
| Dele-GRPO | _69.1_ | 86.8 | 89.0 | _81.2_ | 56.8 | 70.1 | 72.8 | 66.0 | 47.5 | 57.0 | 59.3 | 53.9 | 6.3 |
| Dele-SimKO | **70.1** | _87.5_ | _89.8_ | **82.1** | _57.5_ | _71.2_ | _73.8_ | _66.9_ | _48.1_ | _57.8_ | _60.1_ | _54.6_ | 6.2 |
| TGR-Token | 68.1 | 86.0 | 88.2 | 80.1 | 55.7 | 69.0 | 71.8 | 64.9 | 46.4 | 55.8 | 57.9 | 52.6 | 4.6 |
| TGR-Latent | 68.7 | **88.2** | **91.0** | 80.8 | **57.8** | **72.0** | **74.5** | **67.4** | **48.5** | **58.8** | **61.0** | **55.4** | 5.0 |

## D.2. Additional Token-Space Baselines

Table 11 compares TGR-Latent with ToT and GoT under the same Qwen3-8B setting. Latent-space search provides stronger coverage than token-space branching at matched evaluation budgets.

*Table 11.* **Additional token-space baselines on Qwen3-8B.** Latent-space search outperforms ToT and GoT under matched evaluation settings.

| Benchmark | Method | @1 | @32 | @128 | AUC |
|---|---|---|---|---|---|
| AIME25 | ToT | 22.8 | 30.9 | 33.3 | 28.5 |
| | GoT | 23.7 | 32.1 | 34.5 | 29.7 |
| | TGR-Token | 25.4 | 33.2 | 35.6 | 31.1 |
| | TGR-Latent | 27.8 | 37.4 | 40.1 | 34.8 |
| HumanEval | ToT | 53.9 | 76.9 | 80.4 | 69.0 |
| | GoT | 54.7 | 78.0 | 81.5 | 70.1 |
| | TGR-Token | 55.3 | 78.5 | 81.8 | 70.8 |
| | TGR-Latent | 56.1 | 81.1 | 84.2 | 73.5 |

## D.3. Pilot Open-Ended Transfer

Table 12 evaluates whether latent-space search transfers beyond verifiable math and code tasks. TGR-Latent improves coverage on IFEval and WritingBench while reducing redundancy, as reflected by lower Self-BLEU@32.

*Table 12.* **Pilot open-ended evaluations on Qwen3-8B.** TGR-Latent improves coverage while producing less redundant samples, as reflected by lower Self-BLEU@32.

| Benchmark | Method | @1 | @32 | @128 | AUC | Self-BLEU@32 ↓ |
|---|---|---|---|---|---|---|
| IFEval | Base Model | 49.2 | 72.8 | 79.4 | 64.3 | 79.4 |
| | PSampling | 56.4 | 82.9 | 88.1 | 73.8 | 65.6 |
| | Dele-SimKO | 60.8 | 84.7 | 88.9 | 75.8 | 69.5 |
| | TGR-Token | 57.2 | 84.1 | 89.4 | 75.4 | 59.6 |
| | TGR-Latent | 59.9 | 87.2 | 92.0 | 78.6 | 52.7 |
| WritingBench | Base Model | 18.5 | 54.8 | 66.9 | 47.8 | 78.2 |
| | Dele-SimKO | 26.9 | 69.2 | 77.1 | 59.1 | 66.2 |
| | TGR-Token | 23.6 | 69.8 | 80.6 | 60.0 | 56.1 |
| | TGR-Latent | 25.9 | 73.5 | 84.7 | 63.6 | 48.7 |

## D.4. Capability Breakdown at Fixed Budget

Figure 8 provides the corresponding breakdown for Qwen3-4B; Figure 9 reports a Qwen3-8B capability profile at Pass@128 across benchmarks. A consistent pattern emerges as TGR-Latent maintains strong performance at large $k$, confirming that soft geometric search enhances coverage through effective trajectory diversity. Figure 10 shows that relative Pass@1 gain is stable across scales.

## D.5. Speed–Quality Frontiers Under End-to-End Token Accounting

We compare methods on the *speed–quality* plane: tokens/sec vs AUC. Figure 11 reports frontiers for math (Figures 11a, 11b) and code (Figures 11c, 11d).

## D.6. Method Stability Across Instances

Figure 12 summarizes stability in the AUC–token plane across benchmark instances. A tight spread indicates that improvements reflect systematic robustness rather than outliers.

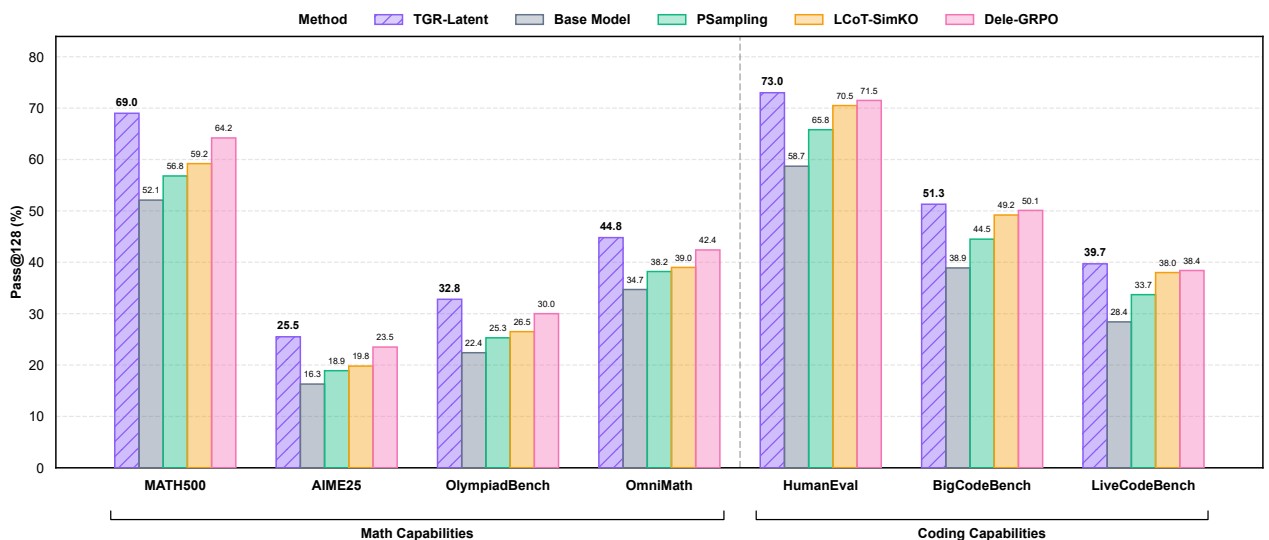

*Figure 8.* **Qwen3-4B capability profile.** Across seven benchmarks, TGR-Latent achieves the highest or near-highest coverage, demonstrating that geometric regularization effectively explores diverse solution modes where token-based sampling saturates.

## E. Latent-Space Geometry Statistics

This appendix presents latent geometry diagnostics, explaining why uniformity $P_{\mathrm{uni}}$ improves robustness beyond chunk structuring. We monitor the geometry of top-scored candidates on MATH500 with Qwen3-8B.

### E.1. Average Pairwise Cosine Similarity (APCS)

At each chunk step $t$, let $\{a_t^{(1)}, \ldots, a_t^{(M)}\}$ denote the top-$M$ candidate anchors ranked by score (with $M = 8$ by default and $M \leq K$). We quantify within-step candidate collapse via the average pairwise cosine similarity:

$$\mathrm{APCS}_t \triangleq \frac{2}{M(M-1)} \sum_{i<j} \frac{a_t^{(i)} \cdot a_t^{(j)}}{\|a_t^{(i)}\|_2 \, \|a_t^{(j)}\|_2}. \tag{25}$$

High APCS indicates near-duplicate top candidates; low APCS indicates random dispersion. A moderate APCS indicates a Goldilocks regime, where candidates remain distinct while maintaining coherence.

### E.2. Effective Candidate-Set Size via Spectral Entropy

Cosine similarity captures redundancy but does not directly measure the effective number of distinct high-scoring alternatives. We therefore estimate an effective candidate-set size $N_{\mathrm{eff}}$ by the entropy of the eigen-spectrum of the covariance of the top-$M$ candidate anchors. Define the covariance

$$\Sigma_t \triangleq \frac{1}{M} \sum_{i=1}^{M} (\tilde{a}_t^{(i)} - \bar{a}_t)(\tilde{a}_t^{(i)} - \bar{a}_t)^\top, \tag{26}$$

where $\tilde{a}_t^{(i)}$ are $\ell_2$-normalized candidate anchors and $\bar{a}_t = \frac{1}{M} \sum_i \tilde{a}_t^{(i)}$. Let $\{\lambda_{t,1}, \ldots, \lambda_{t,M}\}$ be the top-$M$ eigenvalues of $\Sigma_t$ (nonnegative), and normalize them as $p_{t,j} = \lambda_{t,j} / \sum_{m=1}^{M} \lambda_{t,m}$. We define spectral entropy $H_t = -\sum_{j=1}^{M} p_{t,j} \log p_{t,j}$ and convert it to an effective candidate-set size:

$$N_{\mathrm{eff},t} \triangleq \exp(H_t), \qquad N_{\mathrm{eff},t} \in [1, M]. \tag{27}$$

Intuitively, $N_{\mathrm{eff},t} \approx 1$ indicates that the top-scored candidates occupy an almost one-dimensional cone, indicating collapse, while $N_{\mathrm{eff},t} \approx M$ indicates a diverse set of high-scoring alternatives.

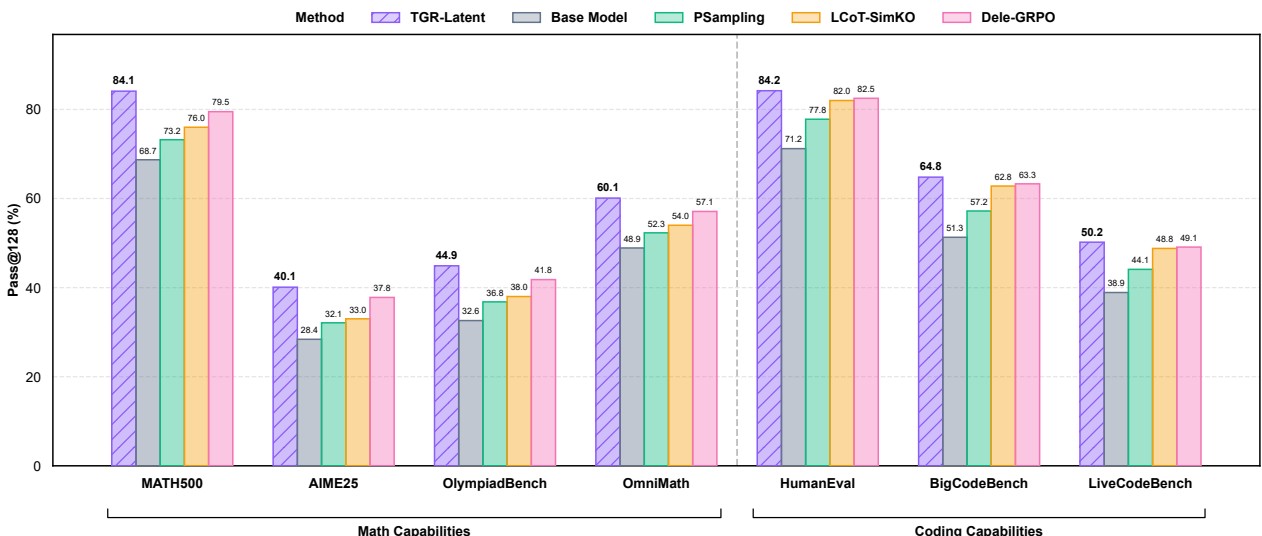

*Figure 9.* **Qwen3-8B capability profile.** Across seven benchmarks, TGR-Latent achieves the highest or near-highest coverage, indicating that latent-space search improves high-$k$ robustness under matched budgets.

### E.3. Cross-Step Dynamics: Forward Momentum and Cycling

We additionally quantify how anchors evolve across steps. The step-to-step similarity $\cos(z_t, z_{t-1})$ measures the directional persistence of the latent state. Extremely high similarity can indicate stagnation, where the search procedure repeatedly re-enters the same latent region. We report a cycling rate, defined as the fraction of steps whose similarity exceeds a high threshold, such as $> 0.99$, as a proxy for repeated revisiting.

### E.4. Results: The Goldilocks Diversity Regime

Table 13 summarizes latent statistics. Removing $P_{\text{uni}}$ collapses candidates into a narrow cone (APCS and $N_{\text{eff}}$ near 1), increasing redundancy and stagnation risk. TGR maintains moderate angular separation and high effective set size, improving coverage without drift. This explains why TGR gains appear primarily in AUC.

*Table 13.* **Latent-space geometry statistics on MATH500 with Qwen3-8B.** This table quantifies how uniformity regularization shapes candidate trajectory structure. Without $P_{\text{uni}}$ search collapses into a narrow latent cone, whereas the full TGR objective sustains diverse and non-redundant exploration.

| Metric | w/o $P_{\text{uni}}$ | TGR (full) | $\Delta$ | Interpretation |
|---|---|---|---|---|
| *Within-step candidate diversity* | | | | |
| Average pairwise cosine similarity (APCS) | $0.96 \pm 0.02$ | $\mathbf{0.68 \pm 0.05}$ | $-0.28$ | Candidate cone expands |
| Minimum pairwise cosine similarity | $0.91 \pm 0.03$ | $\mathbf{0.42 \pm 0.08}$ | $-0.49$ | Presence of outlier trajectories |
| Effective candidate-set size ($N_{\text{eff}}/M$) | $1.2/8.0$ | $\mathbf{6.5/8.0}$ | $+5.3$ | Diverse alternatives |
| *Cross-step trajectory dynamics* | | | | |
| Step similarity $\cos(z_t, z_{t-1})$ | $0.98 \pm 0.01$ | $\mathbf{0.85 \pm 0.04}$ | $-0.13$ | Directional persistence |
| Cycling rate (similarity $> 0.99$) | $34.2\%$ | $\mathbf{4.1\%}$ | $-30.1\%$ | Avoids stagnation |

## F. Efficiency and Systems Notes

This appendix details TGR's efficiency under end-to-end token accounting.

### F.1. End-to-End Overhead Decomposition

Shifting control to inference-time search entails computational costs. For math under the default settings (Table 7), we decompose overhead into lightweight rollouts, scoring, KV cache rebuild, and chunk generation. Table 14 shows that

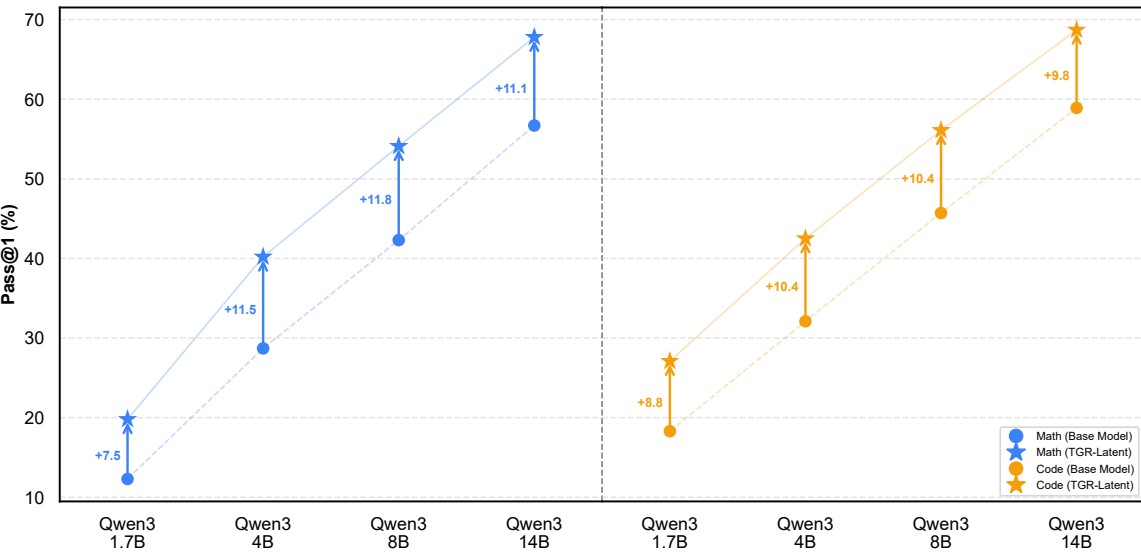

*Figure 10.* **TGR's advantage is scale-invariant.** Relative Pass@1 gains over Base Model remain consistent from 1.7B to 14B on both math and code, confirming that latent-space search provides a fundamental control benefit independent of raw model capacity.

*Table 14.* **Inference overhead decomposition.** Computationally cheap rollouts dominate scoring cost but are amortized via batching. The optimized pipeline adds only $1.11\times$ overhead relative to baseline chunk generation, confirming TGR's budget-efficiency.

| Overhead source | Cost | Optimizable via |
|---|---|---|
| Lightweight rollout | $0.18\times$ | batching and parallelism |
| Candidate scoring | $0.05\times$ | vectorization |
| KV cache reset | $0.03\times$ | boundary-dependent |
| Chunk generation | $1.00\times$ | baseline cost |
| Total (naive) | $1.28\times$ | – |
| Total (optimized) | $1.11\times$ | – |

low-depth rollouts dominate but are amortized.

**Batching and Parallelism**   Rollouts are batchable. We pool $K$ candidate anchors and batch forward passes to amortize kernel launch. This reduces the rollout multiplier from $0.18\times$ to $\sim 0.06\times$ ($s = 32$), lowering total overhead to $1.11\times$. This optimization affects only the execution schedule, not the scoring.

**KV-Cache Reset: Memory Boundedness vs. Rebuild Cost**   Chunk-wise KV reset bounds memory to scale linearly with chunk length $S$ but introduces a rebuild cost. Latency-sensitive deployments can mitigate this by increasing chunk length or triggering resets only when memory limits are reached.

**Candidate Scoring Cost**   Candidate scoring consists of computing the value proxy and geometric penalties, which are implemented as tensorized operations over the batched candidate pool. Empirically, this component contributes only a small fraction of the overhead, consistent with the fact that it avoids autoregressive decoding and is dominated by dense vector operations.

### F.2. Practical Guidance for Throughput-Oriented Serving

For high-throughput serving, two guidelines are particularly effective: batching rollouts across candidates and requests to maximize parallelism, and applying search selectively, for example, only at tool boundaries or when an uncertainty trigger fires. These strategies preserve most of the robustness gains while keeping average latency close to the base model.

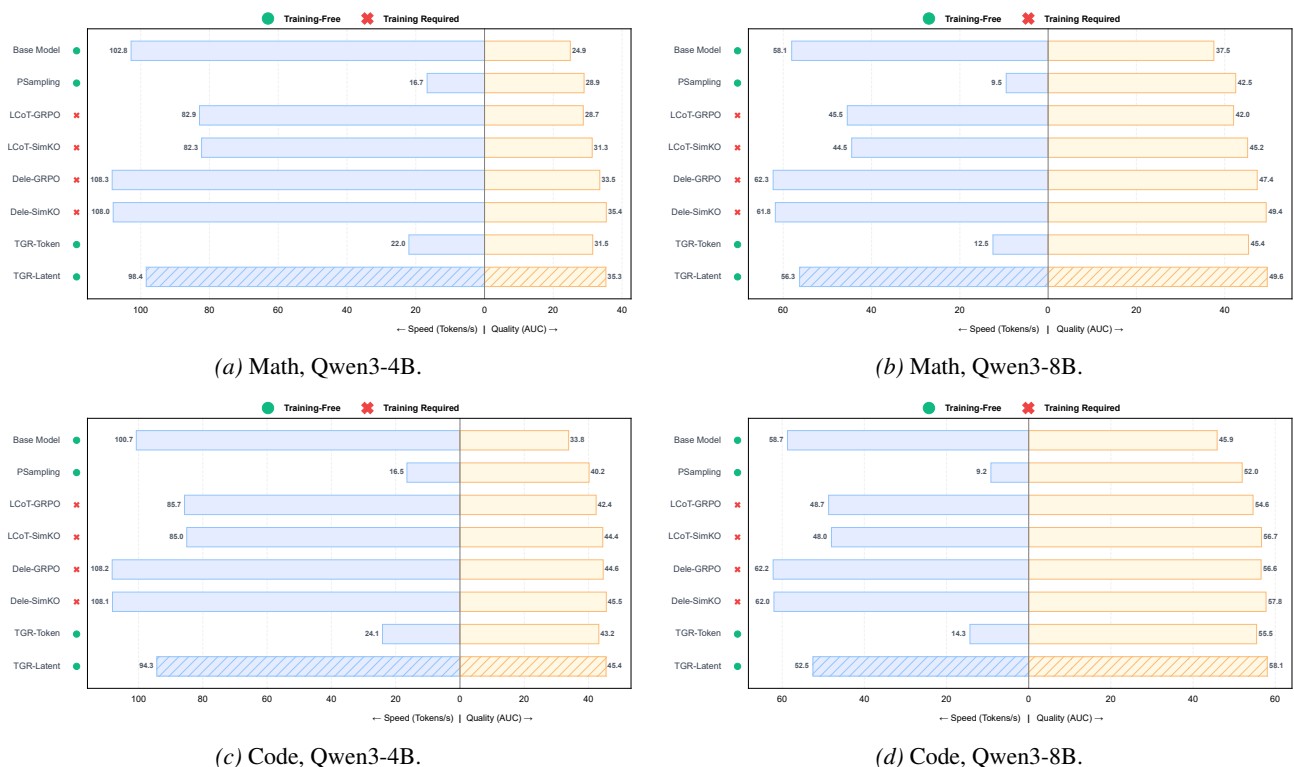

*(a)* Math, Qwen3-4B.  *(b)* Math, Qwen3-8B.

*(c)* Code, Qwen3-4B.  *(d)* Code, Qwen3-8B.

*Figure 11.* **TGR attains the best trade-off between Speed and Quality.** TGR-Latent systematically shifts the frontier upward across domains and model scales, delivering higher robust coverage for the same throughput compared to both training-free sampling and RL-tuned baselines.

### F.3. Chunk-Wise KV Reset and Bounded Context

We reset the KV cache at chunk boundaries to enforce bounded memory, following the bounded-context interface in Appendix B.2 (Eq. 18). We rebuild the cache from the explicit context $Q \oplus \tau_t$ and carry long-horizon information through the injected latent anchor, yielding an $O(S)$ KV-memory bound (Eq. 20) with a small rebuild overhead (Table 14).

### F.4. Greedy Anchor Selection

At each boundary, we execute the single highest-scoring anchor (Eq. 7). Greedy selection keeps overhead low while still benefiting from look-ahead and soft geometric regularization; the selection step itself is negligible compared with rollout cost.

## G. Qualitative Examples and Failure Cases

For each description, we juxtapose short excerpts from four settings: (i) base model, (ii) PSampling, (iii) RL baselines, and (iv) TGR-Latent. Excerpts focus on the decision boundary where trajectories diverge, illustrating the mechanism of improvement or failure.

### G.1. Success Cases

#### Case 1: Escaping a myopic likelihood peak at math.

*Prompt.* Solve for $x$ in the equation $\sqrt{x+6} + \sqrt{2x-1} = 7$ and justify all constraints.

*Base.* Square both sides immediately: $x + 6 + 2x - 1 + 2\sqrt{(x+6)(2x-1)} = 49$, then isolate the radical and square again. After expansion, it obtains a quadratic with candidates but misses domain checks and accepts an extraneous root.

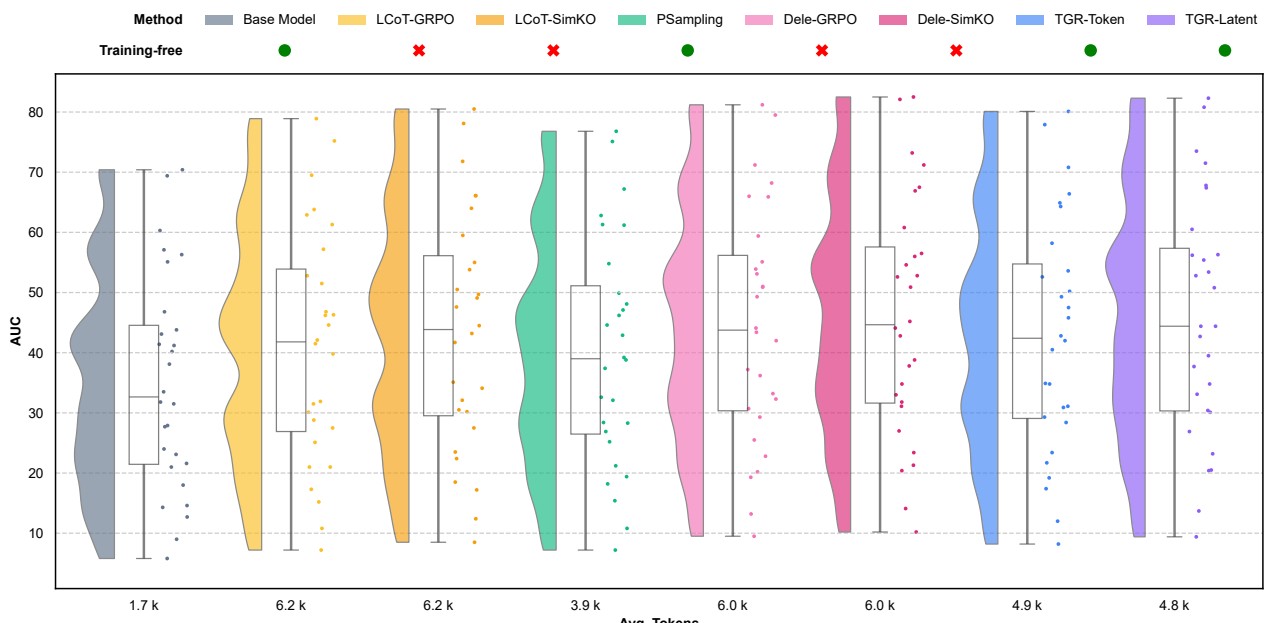

*Figure 12.* **Method stability across benchmark instances.** TGR-Latent clusters in a high-AUC region with compact spread, indicating systematic robustness rather than outlier-driven wins. Green dots mark training-free methods.

*PSampling.* One sample follows the same double-squaring route, another tries substitution $a = \sqrt{x+6}$, $b = \sqrt{2x-1}$ but abandons midway. Overall, most samples cluster around early squaring, and the search budget is spent re-deriving similar algebra with small variations.

*RL baseline.* Commits to an authoritative-looking derivation, adds verbose commentary, but still takes the same early double-squaring path. The trajectory appears confident but narrow, and the final verification step is abbreviated.

*TGR-Latent.* Selects a candidate anchor whose short rollout exposes that domain validation and single-squaring with structured substitution yields a cleaner check: set $a = \sqrt{x+6}$, $b = \sqrt{2x-1}$, note $a + b = 7$ and $a^2 - 2b^2 = 7$. Solve the linear system in $(a, b)$ induced by $a + b = 7$ and $a^2 - 2b^2 = 7$ after expressing $a = 7 - b$, then verify $a \geq 0$, $b \geq 0$ and back-substitute to $x$. Ends with explicit constraint checks, rejecting extraneous candidates.

Lightweight look-ahead down-ranks attractive but failure-prone continuations when they produce unverifiable branches. Soft geometry maintains access to distinct algebraic modes rather than re-sampling duplicates.

---

**Case 2: Maintaining long-horizon intent under bounded context (code).**

*Prompt.* Implement an `LRUCache` supporting `get` and `put` in $O(1)$ time. Use a doubly-linked list plus a hashmap. Edge cases: update existing keys, capacity $= 1$, and repeated `get` should not mutate values.

*Base.* Starts with a correct high-level plan, then, after several screens of code, drifts into an incomplete linked-list update, forgetting to detach nodes cleanly, leading to a stale pointer bug in `move_to_front`.

*PSampling.* Some samples are correct but long, others introduce subtle inconsistencies, for example, updating the value on `get`, or failing to evict the tail sentinel. The diversity is high, but coherence varies sharply, so selecting a correct solution often requires large $N$.

*RL baselines.* Produces a polished implementation with extensive boilerplate. However, it tends to follow a single stylistic pattern, and when that pattern contains a bug, such as wrong eviction order, multiple attempts look similar, making it hard to recover within a fixed budget.

*TGR-Latent.* At a chunk boundary where the base model often starts repeating helper logic, TGR selects an anchor whose rollout emphasizes an invariant-first structure: 1) every node move is `detach` then `attach_front`, 2) hashmap always points to live nodes, 3) eviction is `tail.prev` after detaching. The subsequent chunk implements these primitives cleanly,

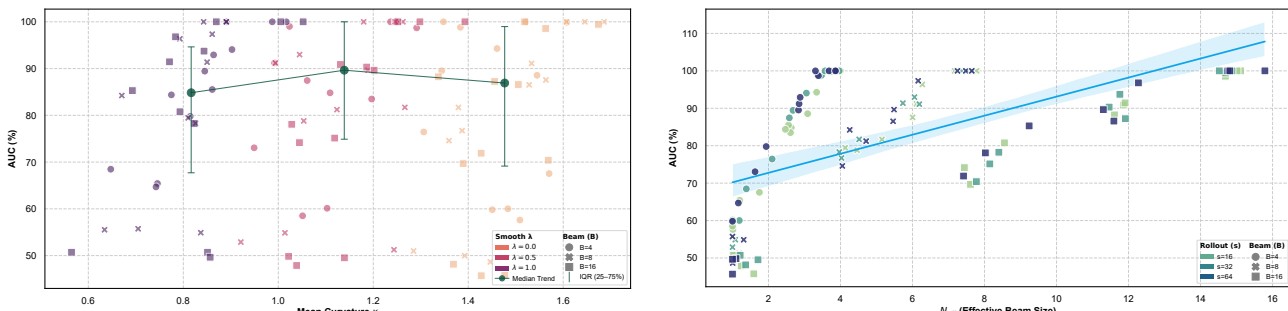

*Figure 13.* **Geometry-Robustness Relationship. Left: Curvature $\kappa$ vs. AUC:** Performance peaks at intermediate curvature, where trajectories balance exploration and coherence. Error bars show IQR across hyperparameter settings. **Right:** $N_{\text{eff}}$ **vs. AUC:** Robustness correlates strongly with effective candidate-set size, supporting the mechanistic claim that TGR's advantage derives from maintaining genuinely distinct search paths rather than redundant variations.

and the final chunk adds targeted tests for the specified edge cases.

Under KV reset, token history is bounded; the anchor carries an intent vector, keeping invariants consistent across chunks. The advantage is not merely higher Pass@1, but fewer wasted branches at high $k$ because trajectories remain diverse yet coherent.

### G.2. Failure Cases

**Case 3: Extremely delayed utility beyond the rollout horizon (math).**

*Prompt.* Prove that for all real $x$, $\ln(1 + x) \leq x$, and characterize when equality holds. Then extend the argument to show $\ln(1 + x) \geq \frac{x}{1+x}$ for $x > -1$.

*Base.* Completes the first inequality via convexity or tangent-line argument, but the extension is shaky, mixes derivatives without a clear plan, and ends inconclusively.

*PSampling.* Eventually finds a clean route by defining $f(x) = \ln(1 + x) - \frac{x}{1+x}$ and analyzing $f'(x)$ and limits, but requires many trials, because intermediate steps do not look immediately rewarding.

*RL baselines.* Often chooses a single confident strategy early, such as Jensen or Taylor, and persists even when it becomes algebraically messy, with limited branching into alternative proofs.

*TGR-Latent.* With lightweight rollouts, several candidate anchors appear equivalently promising, because the discriminative signal (the right auxiliary function and boundary behavior at $x \to -1^+$) only emerges after a longer chain. TGR may select an anchor that produces locally coherent but globally unproductive manipulation, leading to a dead end.

This failure aligns with lightweight look-ahead design: the decisive proof certificate lies beyond the rollout horizon. Mitigation strategies include selective deepening, for example, adaptively increasing $s$ under high uncertainty.

### G.3. Reporting Checklist for Real Logs

To convert these cases into faithful qualitative evidence, we recommend reporting: (i) prompt, (ii) decision boundary chunk, (iii) rollout snippet/score cue (for TGR), and (iv) minimal excerpts showing the critical divergence.

## H. Failure Taxonomy and Practical Mitigations

We summarize common failure modes observed in qualitative analysis (Appendix G) and practical mitigations consistent with TGR's efficiency–accuracy trade-off. Where applicable, diagnostics are grounded in the latent-geometry statistics in Appendix E and sensitivity trends in Appendix C.9.

**Delayed Utility beyond** $s$**.** **Symptom.** Candidates look score-tied under shallow rollouts; selection becomes noisy; errors surface only far downstream. **Mitigation.** Selective deepening: increase $s$ only when score gaps are small or uncertainty spikes; trigger deeper search only at uncertain boundaries (Appendix C.9).

**Over-Regularization.** **Symptom.** High diversity but incoherent branches (too large $\lambda_u$), or overly conservative trajectories (too large $\lambda_b$). **Mitigation.** Small adaptive tuning: increase $\lambda_b$ for syntax-sensitive segments; increase $\lambda_u$ when APCS/$N_{\text{eff}}$ indicate collapse (Appendix E).

**Candidate Collapse and Redundant Exploration.** **Symptom.** Top candidates become highly correlated, and AUC gains saturate early as additional samples yield redundant trajectories. **Diagnostic.** APCS increases and $N_{\text{eff}}$ drops toward 1 (Appendix E). **Mitigation.** Strengthen uniformity (e.g., increase $\lambda_u$ or tighten the repulsion threshold) or modestly increase the candidate budget $K$ to restore effective alternatives, then verify that coherence remains stable.

**Aggressive KV Reset Loss.** **Symptom.** The model forgets exact literals, rare constants, or long instructions that fall outside the bounded explicit context window. **Mitigation.** Retain a short suffix window; reset less frequently; or trigger search only when context truncation is likely to matter (Appendix F).

