# OpenReview forum: "The Geometric Reasoner: Manifold-Informed Latent Foresight Search for Long-Context Reasoning"
_ICML.cc/2026/Conference — ICML 2026 regular_

### Official Review · Reviewer_GWKm · 2026-03-10

**Soundness:** 3
**Presentation:** 2
**Significance:** 3
**Originality:** 3
**Overall Recommendation:** 4
**Confidence:** 1

**Summary:**

This paper presents The Geometric Reasoner (TGR), a training-free inference framework that enhances long-context reasoning via dynamic test-time compute. Unlike conventional methods using discrete token search or heavy RL fine-tuning, TGR works on latent representations at fixed chunk boundaries. It performs manifold-informed search by sampling latent anchors and evaluating them with a geometric scoring function that balances look-ahead utility, trajectory smoothness, and candidate diversity. For scalability, it uses chunk-wise KV cache resets to bound memory at O(S). Experiments on Qwen3 models (1.7B–14B) show improved Pass@k AUC on math and coding benchmarks with only 1.1–1.3× computational overhead.

**Compliance With Llm Reviewing Policy:**

Affirmed.

**Final Justification:**

My concerns have been addressed.

**Key Questions For Authors:**

1.  **Mechanism of Random Injections:** Can you elaborate further on the mechanics behind the fixed, randomly initialized matrices used for anchor extraction and injection? Specifically, why does a random projection of the `<EOC>` token reliably capture the semantic "intent" needed to steer the next chunk without introducing out-of-distribution noise into the residual stream?
2.  **Handling Delayed Rewards:** Regarding the delayed utility failure mode, how sensitive is the algorithm to dynamically scaling the rollout length s? Is there a measurable heuristic or trigger (e.g., high variance in V_fore across candidates) that could be used to adaptively deepen the rollout without exploding the compute budget?
3.  **Future-proofing alongside RL:** Given that the method’s impact is narrowed by RL optimization, how do you envision latent geometric search co-evolving with modern reasoning models that rely heavily on GRPO/PPO? Could the TGR scoring function be integrated directly into the reward modeling phase?
4.  **Chunk Boundary Design:** TGR currently operates on strict, fixed-length chunk boundaries (e.g., S=512). Did you experiment with semantic boundary detection—such as triggering search at line breaks, paragraph ends, or explicit `<thought>` tags—instead of arbitrary token counts?

**Limitations:**

yes

**Strengths And Weaknesses:**

### **Strengths**

* **Novelty and Theoretical Grounding:** Framing test-time compute as a manifold-informed search within a continuous latent space is an elegant departure from discrete token-space search. The insight to use soft geometric regularization instead of hard constraints successfully addresses the "vanishing acceptance rate" issue typical in high-dimensional latent spaces.
* **Memory Efficiency and Practicality:** The systems-level contribution of the chunk-wise KV cache reset is highly practical. By rebuilding the cache from a bounded explicit context, the method bounds peak KV memory to O(S). Combined with batched rollouts, the reported overhead of ~1.11x to 1.3x makes this approach highly viable for real-world deployment.
* **Rigorous Empirical Validation:** The authors provide comprehensive evaluations across multiple model scales (1.7B to 14B) and against strong baselines, including RL-tuned models like LCoT-GRPO and structured state-transition methods like Delethink. Furthermore, the use of normalized AUC over a log-scaled sampling budget is an excellent, robust metric for quantifying trajectory diversity and effective coverage.

### **Weaknesses**

* **Diminished Returns on RL-Tuned Models:** The authors transparently note that TGR yields substantially smaller gains on models that have undergone heavy RL post-training (e.g., Phi-4-reasoning-plus) because RL concentrates the trajectory distribution, shrinking the degrees of freedom for inference-time search. Given the industry's rapid shift toward massive RL post-training for reasoning models, this limits the method's long-term universal applicability.
* **Vulnerability to Delayed Utility Horizons:** Because TGR relies on computationally lightweight, shallow rollouts (e.g., s=32 for math), it struggles when the discriminative signal or "proof certificate" required to evaluate a path lies beyond the rollout horizon. As acknowledged in the failure cases, this can lead to the selection of locally coherent but globally unproductive latent anchors.
* **Justification of Random Projections:** The framework relies on projecting and injecting latent anchors using matrices (W, A_l, B_l) that are initialized randomly (Gaussian) and strictly frozen to maintain a training-free setting. While empirically successful, it is somewhat unintuitive why purely random, untrained low-rank projections can meaningfully and reliably manipulate semantic latent spaces across diverse layers without sometimes catastrophically disrupting the pre-trained manifold.

---

> ### Author Rebuttal · Authors · 2026-03-29
>
> We thank the reviewer for highlighting TGR as a simple, training-free, architecture-agnostic way to turn modest test-time compute into stronger robust coverage under bounded memory.
>
> **RL and post-training**
>
> Heavy RL can reduce the marginal room left for inference-time search as concluded in our manuscript, but this may not make TGR less relevant in the long run. This interpretation is also consistent with recent RLVR evidence suggesting that post-training mainly reallocates probability mass toward higher-reward reasoning paths already present in the base model, rather than reliably creating fundamentally new reasoning patterns, and that this can narrow multi-sample diversity [1]. These make TGR complementary to RL rather than obsolete under it. A viable approach may involve leveraging our manifold strategy as an auxiliary post-training signal that enhances reasoning while preserving the trajectory distribution.
>
> **Delayed utility beyond the rollout horizon**
>
> Delayed reward is indeed a genuine challenge for lightweight look-ahead methods. Nevertheless, Table 3 shows that shallow foresight ($V_{{fore}}$) is still the dominant contributor, being imperfect yet sufficiently informative. The natural adaptation is selective deepening rather than globally increasing $s$. When score gaps are small or uncertainty spikes, one can re-evaluate only the top few candidates. Table 7 supports exactly this direction: Top-2 re-evaluation reaches 53.5 Avg. AUC at about $1.14\times$ overhead, and $N_{\text{MC}}=2$ reaches 53.3 at about $1.17\times$.
>
> **Random Latent Interface**
>
> The semantic content resides entirely in the top-layer hidden state of the \<EOC\> token itself, not in the random matrices. The matrix $W$ merely establishes a fixed coordinate system. In fact, when $d_z = d_h$, $W$ can simply be the identity matrix (Appendix A), proving that no privileged learned projector is required. The random injection does not overwrite or disrupt the pretrained residual stream. Because the same frozen, low-rank interface is used consistently during both rollout evaluation and final chunk generation, it acts as a stable, self-consistent perturbative bias. This allows the model to reliably compare and condition on local trajectory variations in a fixed coordinate space, rather than forcing it to interpret a newly learned semantic channel from scratch. Empirical ablations confirming that performance is driven by this score-guided selection rather than arbitrary random perturbation are detailed in our response to Reviewer YguA.
>
> **Chunk boundary design**
>
> Using fixed-length boundaries is a deliberate mathematical and systems design choice. Mathematically, TGR's geometric penalties are defined over a discrete latent trajectory, where $P_{\text{bum}}$ computes a second-order difference across hidden states. A fixed chunk length $S$ provides a strict equidistant discretization, ensuring that curvature and smoothness penalties remain mathematically rigorous and comparable across all steps. Relying on semantic boundaries would introduce nonuniform step sizes, making the geometric penalty less directly comparable across steps. It would also weaken the predictability of chunk-wise KV resets and reduce the efficiency of batched candidate evaluation. While we do agree semantic boundaries are promising, fixed-length chunking provides the necessary mathematical consistency and hardware-friendly foundation required for the current framework.
>
> ---
>
> [1] Yue Y, et al. "Does reinforcement learning really incentivize reasoning capacity in llms beyond the base model?." arXiv preprint arXiv:2504.13837 (2025).

---

> > ### Author Rebuttal · Reviewer_GWKm · 2026-04-02
> >
> > Thanks for your response. I maintain my score.

---

### Official Review · Reviewer_YguA · 2026-03-11

**Soundness:** 2
**Presentation:** 3
**Significance:** 3
**Originality:** 4
**Overall Recommendation:** 4
**Confidence:** 2

**Summary:**

This paper proposes The Geometric Reasoner (TGR), a training-free inference-time framework for long-context reasoning that performs search in a latent anchor space rather than directly in token space. The method segments generation into chunks, extracts a chunk-level latent state at each boundary, samples candidate anchors in its local neighborhood, and scores them using a combination of lightweight look-ahead rollout value, a smoothness-oriented bumpiness penalty, and a diversity-promoting uniformity term. The selected anchor is then injected back into the frozen backbone to guide the next chunk, while chunk-wise KV cache resets keep memory bounded linearly in chunk length. Experiments on math and code benchmarks show that TGR improves trajectory coverage, primarily measured by Pass@k AUC, over sampling-based and training-based baselines under matched end-to-end token budgets, with modest reported overhead.

**Compliance With Llm Reviewing Policy:**

Affirmed.

**Final Justification:**

My concerns have been adequately addressed, thus I maintain my positive score.

**Key Questions For Authors:**

See weaknesses.

**Limitations:**

yes

**Strengths And Weaknesses:**

## Strengths

1. **The paper studies an important problem setting:** improving long-context reasoning at test time without additional training and under bounded memory. The combination of chunk-wise search, latent anchor selection, and KV-cache reset is targeted at a practically relevant regime where naive long-horizon exploration is expensive.

2. **The proposed method is reasonably well-motivated and technically coherent.** TGR performs search over chunk-level latent anchors, scores candidates with a combination of short-horizon foresight, bumpiness regularization, and diversity regularization, and then conditions generation through a lightweight residual injection interface. This makes the method more structured than plain sampling while remaining training-free.

3. **The empirical evaluation is fairly broad.** The paper evaluates on multiple math and code benchmarks, reports results across several Qwen3 scales, and includes comparisons to both sampling-based and RL-based baselines.

4. **The ablation and diagnostic sections are helpful.** The component ablation suggests that foresight is the main driver, while the uniformity and bumpiness terms provide complementary gains. The latent-geometry diagnostics and candidate-diversity analysis also help clarify the intended mechanism behind the AUC improvements.


## Weaknesses

1. **The main empirical wins are concentrated on AUC / high-k robustness rather than clearly stronger single-trajectory performance.** The paper itself argues that the advantage appears mainly in AUC rather than Pass@1, which makes the practical significance somewhat less clear for settings where only one or a small number of samples are used.

2. **Variance reporting is insufficiently clear given the reliance on large-k metrics such as Pass@128.** The paper shows error bands / “±1σ” in the hyperparameter study, but it does not clearly specify how many repeated runs or random seeds these statistics come from. This matters because high-k estimates can be sensitive to sampling noise, and the credibility of the reported AUC improvements would be stronger with explicit repeat counts and confidence intervals.

3. **Some of the central design choices remain fairly heuristic.** In particular, the latent control interface relies on fixed random projection/injection matrices rather than learned or semantically validated control variables. While this keeps the method training-free, it also makes it harder to judge how much of the gain comes from the proposed geometric interpretation versus from a more generic structured perturb-and-rank procedure.

4. **The baseline set still leaves a few important gaps.** The paper discusses self-consistency/ToT/GoT-style methods conceptually and introduces TGR-Token as a token-space control ablation, but there is no explicit vanilla self-consistency baseline in the main results. That makes it harder to isolate how much improvement comes from latent-space search specifically versus simply spending compute on additional structured sampling.

---

> ### Author Rebuttal · Authors · 2026-03-29
>
> We thank the reviewer for their time and constructive feedback, and for engaging directly with whether training-free latent search improves long context reasoning under matched end-to-end budgets.
>
> **AUC and high-$k$ robustness**
>
> AUC and medium/high-k performance are informative metrics for evaluating whether additional inference-time compute is actually converted into additional useful trajectories rather than near-duplicates under unified token accounting. Main results show that TGR sustains marginal gains beyond $k=32$ while remains competitive at $k=1$. The practical value of our method can be pronounced when integrated into modern reasoning systems, where improving the candidate distribution yields practical benefits for multi-sample generation, validation, and screening.
>
> **Variance reporting**
>
> The $\pm 1\sigma$ error bands represent the sample standard deviation across 3 independent runs with seeds 42, 1234, and 3407. In each run, AUC is computed over the full log-spaced sampling grid $k \in \\{1,2,4,8,16,32,64,128\\}$. We will make this repeat protocol explicit in the revision.
>
> **Heuristic latent interface**
>
> The fixed random interface is heuristic, but not vacuous. It is designed to provide a stable control variable for local perturbation, rather than to encode interpretable semantic coordinates. The semantic content resides in the model-produced chunk-end hidden state, while directional control arises from score-guided anchor selection through the fixed injection interface. The key question is whether this setup supports meaningful control. Table 3 shows that it does: replacing the selected anchor with a random anchor drops Avg. AUC from 53.2 to 38.3, removing $V_{\text{fore}}$ drops it to 43.7, and TGR-Latent remains above TGR-Token at 48.8. Figure 7 further shows that selected anchors consistently have higher $V_{\text{fore}}$ than rejected ones, while Table 7 shows that the gains are stable across changes in $r$, $d_z$, $\eta$, and rollout strategy. These suggest that benefit comes from score-guided latent search that preserves effective independence, rather than chunking alone or arbitrary perturbation. For a mechanistic account of how a fixed random interface supports stable control, please see our response to Reviewer GWKm.
>
> **Baseline scope and token-space baselines**
>
> We prioritized the baselines central to the comparison and included TGR-Token as the most direct token-space counterpart under the same conditions. Vanilla self-consistency is orthogonal to ToT / GoT / TGR-Token. It is a post-hoc aggregation rule over completed samples rather than an inference-time search procedure. However, we agree that adding ToT and GoT makes the distinction between token-space and latent-space search more explicit. In the added Qwen3-8B results below on AIME25 and HumanEval, ToT and GoT remain below TGR-Token. GoT comes closer on code tasks because code generation tasks admit more modular, multi-path solution structures and are therefore more amenable to graph-structured branching and exploration. We will include the full results across all benchmarks in the revised appendix.
>
> | AIME25     | @1   | @32  | @128 | AUC  |
> | ---------- | ---- | ---- | ---- | ---- |
> | ToT        | 22.8 | 30.9 | 33.3 | 28.5 |
> | GoT        | 23.7 | 32.1 | 34.5 | 29.7 |
> | TGR-Token  | 25.4 | 33.2 | 35.6 | 31.1 |
> | TGR-Latent | 27.8 | 37.4 | 40.1 | 34.8 |
>
> | HumanEval  | @1   | @32  | @128 | AUC  |
> | ---------- | ---- | ---- | ---- | ---- |
> | ToT        | 53.9 | 76.9 | 80.4 | 69.0 |
> | GoT        | 54.7 | 78.0 | 81.5 | 70.1 |
> | TGR-Token  | 55.3 | 78.5 | 81.8 | 70.8 |
> | TGR-Latent | 56.1 | 81.1 | 84.2 | 73.5 |

---

> > ### Author Rebuttal · Reviewer_YguA · 2026-04-01
> >
> > Thanks for your detailed response. I maintain my positive score.

---

### Official Review · Reviewer_MK1M · 2026-03-13

**Soundness:** 3
**Presentation:** 3
**Significance:** 2
**Originality:** 2
**Overall Recommendation:** 3
**Confidence:** 3

**Summary:**

paper introduces a framework called TGR designed to improve how AI models handle long, complex reasoning tasks without needing extra training. The authors point out that when models think through difficult problems, they often waste effort on repetitive ideas or get lost in irrelevant details. To solve this, TGR uses a "foresight search" that looks ahead at potential paths in the model's hidden mathematical space. It uses a clever "chunk-based" approach where it evaluates different directions at specific intervals and uses geometric rules to ensure the model’s "train of thought" remains smooth, diverse, and focused on the most promising solution.

**Compliance With Llm Reviewing Policy:**

Affirmed.

**Key Questions For Authors:**

whether TGR can be adapted for real-time applications, such as live coding assistants, where the delay from the look-ahead search might be more noticeable to a user.

**Limitations:**

the "candidate collapse" problem mentioned in the appendix, where the top choices can become too similar over time, potentially limiting the model's creativity in very long sessions. Another limitation is that the framework requires a high degree of "internal transparency" from the model, meaning it might not work with "closed-box" AI systems where we cannot see or manipulate the hidden mathematical layers. Lastly, the authors should clarify if the performance gains remain consistent as the context window grows to millions of tokens, or if the "geometric" calculations eventually become a bottleneck.

**Strengths And Weaknesses:**

strength: efficiency and its ability to significantly boost performance on hard reasoning benchmarks like mathematical proofs and complex coding. Because it is a "training-free" method, it can be applied to existing models immediately without the massive cost of retraining them. The use of "latent anchors" is a standout feature, as it allows the model to summarize its previous thoughts into compact representations, saving a huge amount of memory. Furthermore, the introduction of a "soft geometric regularizer" is a brilliant way to prevent the model from becoming redundant, ensuring that different search paths actually explore different ideas rather than just repeating the same logic in different words. The results show that TGR outperforms standard search methods like "Best-of-N" while using much less memory and time.


weak: the method is highly sensitive to the "chunk size" chosen for the search; if the chunks are too small, the model becomes too slow, but if they are too large, the foresight search might miss critical turning points in the logic. Second, the "manifold-informed" part of the logic assumes that the model's hidden layers are well-organized, but for smaller or less-optimized models, these mathematical spaces might be too "noisy" for the geometric rules to work effectively. Third, the framework currently relies on a specific "KV cache" management system to save memory, which makes it technically difficult to implement on diverse hardware setups that use different memory optimization tricks. Finally, while the paper excels at math and logic, it does not explore how well this "geometric" approach works for more subjective or creative tasks where there isn't one clear "correct" manifold path to follow.

---

> ### Author Rebuttal · Authors · 2026-03-29
>
> We thank the reviewer for recognizing the practical value of training-free long-horizon reasoning under bounded memory.
>
> **Chunk size, latency, and real-time serving**
>
> Fixed chunking introduces a deliberate latency–adaptivity trade-off in the training-free interface rather than a brittle dependency. The true question is whether TGR is highly sensitive to this choice, and our results suggest it is not. As detailed in Appendix A, TGR uses a fixed <EOC> position to ensure consistent anchor extraction across tasks and stopping behaviors. Since $P_{\text{bum}}$ is a discrete second-order difference on the latent trajectory, a fixed $S$ provides a uniform discretization that keeps the bumpiness term strictly comparable across steps. Empirically, Figure 4 shows smooth gains with diminishing returns as rollout depth $s$ and candidate budget $K$ increase, while Table 11 reports only $1.11\times$ optimized overhead. Thus, TGR shows promise for real-time applications, but it is more suited for high-value, long-horizon reasoning tasks than for keystroke-level serving. For the more geometric rationale behind fixed chunk boundaries, please see our response to Reviewer GWKm.
>
> **Smaller models and Collapse**
>
> For smaller models, the TGR interface still preserves a useful local ordering among candidate anchors. At each boundary, it perturbs only the local neighborhood of the previous anchor, then ranks candidates with the soft score, where lightweight foresight $V_{\text{fore}}$ provides the main separation signal. Figure 7 shows that selected anchors consistently have higher $V_{\text{fore}}$ than rejected ones, indicating that the interface induces a meaningful local ranking rather than arbitrary perturb-and-rank behavior. Empirically, Tables 8 and 9 show that TGR-Latent remains effective from Qwen3-1.7B through 14B.
>
> Candidate collapse is a common redundancy pathology in multi-sample reasoning, and preventing it is exactly why we introduced TGR's uniformity regularizer $P_{\text{uni}}$. Appendix H is intended as a failure-mode and troubleshooting guide. It describes the symptoms that emerge when the uniformity penalty is disabled or set too low, such as top-ranked anchors becoming overly correlated and cycling within a narrow cone. Under TGR's default configuration, this behavior is explicitly mitigated. Table 10 shows that adding $P_{\text{uni}}$ lowers APCS from 0.96 to 0.68, raises $N_{\text{eff}}$ from 1.2/8.0 to 6.5/8.0, and reduces cycling from 34.2% to 4.1%. Table 3 shows the corresponding AUC drop when $P_{\text{uni}}$ is removed.
>
> **KV cache, hardware stack, and million-token contexts**
>
> TGR does not depend on a particular cache manager. It changes the inference interface by resetting the cache at chunk boundaries and rebuilding only from a bounded explicit context $Q \oplus \tau_{t-1}$, yielding an $O(S)$ KV-memory bound independent of total horizon. Compression, merging, or selective-retention methods can still be applied inside the bounded window that TGR keeps. They are complementary rather than conflicting.
>
> The same decomposition explains that in million-token scenarios, the explicit KV bound remains $O(S)$. Table 11 further shows that the main overhead comes from lightweight rollouts (which are batch-amortizable), while candidate scoring constitutes a minor fraction (and is vectorizable), leading to a total optimized multiplier of $1.11\times$. Therefore, geometry computations are not expected to be the bottleneck in ultra-long context deployments.
>
> **Closed-box and open-ended tasks**
>
> Long-context reasoning is a general challenge across model settings. TGR is most directly applicable to open-weight, local, or provider-internal deployments where latent states are accessible, which makes these settings the natural place to validate the method. For broader open-ended evaluation, we provide additional results in our response to Reviewer 5BBR.

---

> > ### Author Rebuttal · Reviewer_MK1M · 2026-04-03
> >
> > The reply is fairly simple and most my concerns were not addressed. I choose to remain the current score

---

> > > ### Author Response · Authors · 2026-04-05
> > >
> > > We thank the reviewer for the continued engagement. For clarity, we respond point by point below and add a new direct sweep for chunk-size sensitivity.
> > >
> > > **1. Chunk size sensitivity.**
> > >
> > > To isolate chunk length without conflating it with the total reasoning budget, we vary $S \in \\{256, 512, 1024\\}$ and adjust the maximum chunks $L$ so the horizon remains strictly fixed at $L \times S = 12288$. All other settings remain exactly as in Table 6 (Qwen3-8B, $K = 8$, $s = 32$ for math, $s = 64$ for code).
> > >
> > > | Benchmark | Chunk Length ($S$)  | AUC  | Pass@128 | Overhead     |
> > > | :-------- | :------------------ | :--- | :------- | :----------- |
> > > | AIME25    | $S = 256$           | 35.2 | 40.7     | 1.17$\times$ |
> > > |           | $S = 512$ (default) | 34.8 | 40.1     | 1.11$\times$ |
> > > |           | $S = 1024$          | 33.9 | 38.9     | 1.05$\times$ |
> > > | HumanEval | $S = 256$           | 73.7 | 84.5     | 1.16$\times$ |
> > > |           | $S = 512$ (default) | 73.5 | 84.2     | 1.11$\times$ |
> > > |           | $S = 1024$          | 72.8 | 83.1     | 1.06$\times$ |
> > >
> > > This sweep shows a predictable efficiency-adaptivity trade-off rather than brittleness. Smaller chunks improve control frequency at higher overhead, while larger chunks reduce overhead but mildly reduce coverage. The default $S=512$ remains near the best operating point.
> > >
> > > **2. Real-time latency.**
> > >
> > > Figure 4 shows diminishing returns as rollout depth $s$ and candidate budget $K$ increase, indicating a stable operating regime. Appendix B.1 gives the runtime dependence, and Table 11 shows only $1.11\times$ optimized overhead because lightweight rollouts are batch-amortizable. TGR is therefore practical for long-horizon reasoning with bounded added latency, while being more suited to higher-value long-horizon settings than to keystroke-level autocomplete.
> > >
> > > **3. Noisy latent space in smaller models.**
> > >
> > > Figure 7 shows selected anchors consistently have higher foresight value ($V_{\text{fore}}$) than rejected ones, indicating a meaningful and stable local ranking signal. Tables 8-9 show TGR-Latent remains effective from Qwen3-14B to Qwen3-1.7B, and Figure 10 shows stable relative Pass@1 improvement across scales.
> > >
> > > **4. KV cache hardware compatibility.**
> > >
> > > Appendix B.2 shows that rebuilding from a bounded explicit context yields an algorithmic $O(S)$ KV-memory bound. TGR is therefore an inference interface rather than a hardware-specific cache manager, and hardware-level cache compression or selective-retention methods are complementary to it.
> > >
> > > **5. Creative and subjective tasks.**
> > >
> > > To address applicability beyond math and code, we include two pilot open-ended evaluations on IFEval and WritingBench (Literature & Arts), as in our response to Reviewer 5BBR. Here, we present the key results directly. In both settings, TGR-Latent achieves the highest AUC and the lowest Self-BLEU@32 among the reported methods. These results indicate improved quality together with substantially greater diversity.
> > >
> > > | IFEval     | AUC  | Self-BLEU@32 $\downarrow$ |
> > > | ---------- | ---- | ------------------------- |
> > > | Dele-SimKO | 75.8 | 69.5                      |
> > > | TGR-Token  | 75.4 | 59.6                      |
> > > | TGR-Latent | 78.6 | 52.7                      |
> > >
> > > | Literature & Arts | AUC  | Self-BLEU@32 $\downarrow$ |
> > > | ----------------- | ---- | ------------------------- |
> > > | Dele-SimKO        | 59.1 | 66.2                      |
> > > | TGR-Token         | 60.0 | 56.1                      |
> > > | TGR-Latent        | 63.6 | 48.7                      |
> > >
> > > **6. Candidate collapse in long sessions.**
> > >
> > > This is the redundancy pathology targeted by $P_{\mathrm{uni}}$. Table 10 shows that adding $P_{\text{uni}}$ reduces APCS from 0.96 to 0.68, raises effective candidate-set size from 1.2 to 6.5 (out of 8.0), and reduces cycling from 34.2% to 4.1%. Table 3 shows the accompanying AUC drop when this term is removed. Candidate collapse is therefore an explicitly diagnosed and resolved pathology in our method.
> > >
> > > **7. Closed-box inapplicability.**
> > >
> > > TGR is implemented at the inference-interface level through anchor extraction and residual injection, which require access to internal hidden states. Open-weight, local, and provider-internal deployments are therefore the settings in which the method is directly implementable and evaluated in this paper. They provide a direct setting for empirical validation rather than a restriction to open-weight models.
> > >
> > > **8. Million-token context scalability.**
> > >
> > > Appendix B.2 shows that explicit KV-memory remains bounded at $O(S)$ because older history is transmitted through the low-dimensional latent anchor rather than retained in cache. Table 11 further shows candidate scoring is a minor tensorized cost, so geometric computation does not become the dominant systems bottleneck.
> > >
> > > ---
> > >
> > > We are happy to clarify any further technical issues. Additional concrete technical feedback would be especially valuable in helping us improve the work and strengthen the paper.

---

### Official Review · Reviewer_5BBR · 2026-03-16

**Soundness:** 3
**Presentation:** 3
**Significance:** 3
**Originality:** 4
**Overall Recommendation:** 5
**Confidence:** 1

**Summary:**

The authors propose TGR, a inference time algorithm for efficiently allocating compute to search diverse trajectories instead of ending up with redundant ones. Compared to training-free test-time compute baselines, RL-tuned approaches (that internalize long CoT), and RL with structured state transitions, the proposed method achieves better performances while also balancing the total number of generated tokens.

**Compliance With Llm Reviewing Policy:**

Affirmed.

**Key Questions For Authors:**

see weaknesses above

**Limitations:**

yes

**Strengths And Weaknesses:**

* The analyses in section 5 are interesting, showing that how theory could lead to better empirical results.
* In addition to Qwen3, the authors also test the results using Phi-4, which checks that the approach works in different model families as well.
* The component analyses in Table 3 checks that all components contribute to the performance gain and that V_fore is very important, which checks the validity of the proposed approach

As for weaknesses,
* Can you provide results beyond math problems and code generation?

---

> ### Author Rebuttal · Authors · 2026-03-29
>
> We thank the reviewer for the positive assessment and for encouraging us to evaluate beyond math and code.
>
> **Results on open-ended tasks**
>
> We used math and code as the main testbeds because they combine long context reasoning, strict objective verification, and standardized Pass@k/AUC evaluation under matched end-to-end forward-token accounting.
>
> We added two pilot open-ended experiments on IFEval [1] and WritingBench* (Literature & Arts subset) [2] with Qwen3-8B. For these additional domains, we also report Self-BLEU@32, because the question here is not only whether samples are good, but also whether they are genuinely different under a fixed sampling budget. The tables below show that the latent-space search mechanism transfers beyond math and code as well. We will include the full set of results in our revision.
> | IFEval     |   @1 |  @32 | @128 |  AUC | Self-BLEU@32 $\\downarrow$ |
> | :--------- | :---: | :---: | :---: | :---: | :------------------------: |
> | Base Model | 49.2 | 72.8 | 79.4 | 64.3 |  79.4 |
> | PSampling  | 56.4 | 82.9 | 88.1 | 73.8 |   65.6 |
> | Dele-SimKO | 60.8 | 84.7 | 88.9 | 75.8 | 69.5 |
> | TGR-Token  | 57.2 | 84.1 | 89.4 | 75.4 |  59.6 |
> | TGR-Latent | 59.9 | 87.2 | 92.0 | 78.6 |  52.7 |
>
> | WritingBench* | @1   | @32  | @128 | AUC  | Self-BLEU@32 $\\downarrow$ |
> | :----------------- | :----: | :----: | :----: | :----: | :--------------: |
> | Base Model        | 18.5 | 54.8 | 66.9 | 47.8 | 78.2 |
> | Dele-SimKO        | 26.9 | 69.2 | 77.1 | 59.1 | 66.2 |
> | TGR-Token         | 23.6 | 69.8 | 80.6 | 60.0 | 56.1 |
> | TGR-Latent        | 25.9 | 73.5 | 84.7 | 63.6 | 48.7 |
>
> ---
>
> [1] Zhou J, et al. "Instruction-following evaluation for large language models." arXiv preprint arXiv:2311.07911 (2023).
>
> [2] Wu Y, et al. "Writingbench: A comprehensive benchmark for generative writing." arXiv preprint arXiv:2503.05244 (2025).

---

### Decision · Program_Chairs · 2026-04-30

**Decision:**

Accept (regular)

**Comment:**

TGR is a training-free framework for long-context reasoning that performs manifold-informed search over latent anchors at chunk boundaries, combining lightweight foresight with soft geometric regularizers for smoothness and diversity under bounded memory. Reviewers praised the elegant latent-space search formulation, strong Pass@k AUC gains across model scales, and practical memory efficiency. The rebuttal convincingly addressed concerns around chunk-size sensitivity, the random injection interface, baseline coverage, and generalization beyond math and code via open-ended evaluations on IFEval and WritingBench. I recommend acceptance.